# Profiles of social isolation and loneliness as moderators of the longitudinal association between uncorrected hearing impairment and cognitive aging

Charikleia Lampraki [1,2,3] ✉, Sascha Zuber[2,3], Nora Turoman[2], Emilie Joly-Burra[1,2,3], Melanie Mack [2,3], Gianvito Laera[1,2,3,4], Chiara Scarampi[2,3], Adriana Rostekova[1,2,3], Matthias Kliegel[1,2,3] & Andreas Ihle[1,2,3]

Hearing impairment affects a growing number of older adults and is linked to cognitive decline. This study investigated whether profiles of social isolation and loneliness (e.g., non-isolated but lonely) moderate the association between hearing impairment and cognition over time across domains. Using longitudinal data from waves 1–9 of the Survey of Health, Ageing, and Retirement in Europe (SHARE), we analysed 33,741 individuals (Mage = 61.4, SD = 8.6) using multilevel models accounting for both inter-and intra-individual variability. Results showed that both higher levels and worsening self-reported hearing impairment were associated with lower cognitive performance and steeper decline in episodic memory (immediate and delayed recall) and executive functioning (verbal fluency). Notably, profiles combining social isolation and/or loneliness were linked to lower cognitive performance across domains. Furthermore, for the "non-isolated but lonely" profile hearing impairment was more strongly and negatively associated with episodic memory decline compared to the non-isolated and not lonely profiles. A separate multivariate model confirmed that the moderating role of social isolation and loneliness profiles differed across cognitive domains. Specifically, among individuals in the non-isolated but lonely group, the negative association between hearing impairment and cognition was strongest for episodic memory compared to executive functions. These findings underscore the importance of considering both sensory and psychosocial factors in cognitive aging. Addressing hearing impairment alongside loneliness—even in socially integrated individuals—may be crucial for promoting cognitive health in later life.

Hearing impairment, ranging from partial to total hearing loss, affects a significant proportion of older adults, with prevalence increasing with age. According to the World Health Organization (WHO)[1], approximately 65% of adults aged 60 and older experience hearing loss, with one in four affected at a moderate to severe level. Hearing impairment relates to adverse outcomes in various domains of life, such as cognition[2–4], mental health[5,6], and quality of life[7]. For example, Livingston et al.[2]. estimated an increased risk for dementia of 90% for older adults who are impaired in their hearing capacity compared to the non-impaired. With increasing longevity, understanding the factors that are associated with variation in the relationship between hearing impairment and cognitive functioning is, therefore, of utmost importance for aging societies. The link between hearing impairment (objective and subjective) and cognitive functioning has been repeatedly observed in prior research[3,4,8–10] – suggesting that sensory decline relates to poorer cognitive outcomes. Yet, the relationship between hearing loss and cognition is not uniform across individuals.

However, there is still a lack of fine-grained research examining factors that may be associated to stronger or weaker associations between hearing

[1]Faculty of Psychology and Educational Sciences, University of Geneva, Geneva, Switzerland. [2]Center for the Interdisciplinary Study of Gerontology and Vulnerability, University of Geneva, Geneva, Switzerland. [3]Swiss Center of Expertise in Life Course Research LIVES, Geneva, Switzerland. [4]University of Applied Sciences and Arts Western Switzerland HES-SO, Geneva School of Health Sciences, Geneva Musical Minds Lab (GEMMI Lab), Geneva, Switzerland. ✉e-mail: charikleia.lampraki@unige.ch

impairment and cognitive aging. While hearing impairment is widely recognized as a risk factor for cognitive decline, not all individuals with hearing loss experience the same cognitive trajectory. This suggests that other factors, such as social isolation and loneliness, may be associated with differences in the strength of this relationship, potentially explaining inter-individual differences in cognitive resilience. However, research investigating which factors are linked to amplified or attenuated associations between hearing impairment and cognitive outcomes remains limited.

Social isolation and loneliness are particularly relevant in this context, as they often increase with age. Older adults commonly experience life transitions that tend to diminish the size and quality of their social network, such as retirement and bereavement[11], often resulting in social isolation and feelings of loneliness. Social isolation represents an objective state in which the individual has very few or none to interact with in a frequent manner, while loneliness, on the other hand, is the subjective feeling of being isolated, and stems from the imbalance between the need for socialization and the lack of it[12]. In the USA[12], it is estimated that approximately 25% of older aged individuals is socially isolated, while in Europe the prevalence is around 22%[13]. Regarding loneliness, a large metanalysis reported that in Europe loneliness ranges between 6.5% and 24.2% for older adults, depending on the geographical region[14]. Social isolation has been associated to poorer mental health and cardiovascular outcomes[15], while loneliness can vary depending on the duration and context of the experience (i.e., situational loneliness, chronic loneliness), and has been linked to a higher risk of all-cause mortality[16]. Importantly, Holt-Lundstad and colleagues[17] emphasize in their metanalysis that individuals with limited social relationships or non-satisfying ones show higher mortality risk, which can be compared in magnitude to that of tobacco use, highlighting the public health implications.

Social isolation and loneliness represent distinct phenomena, and have been investigated concurrently in the past as the objective and subjective form of isolation[18]. However, rather than treating social isolation and loneliness as independent risk factors, Menec et al.[19,20] emphasize that these experiences should be analysed together as distinct psychosocial profiles. Their framework categorizes individuals into four profiles: (a) non-isolated and not lonely, (b) non-isolated but lonely (lonely-in-the-crowd), (c) isolated but not lonely, and (d) both isolated and lonely. We have adopted this approach in the present study since it offers a framework that is better reflecting the actual social circumstances of individuals and how they associate to cognitive aging. They discuss[19,20], in line with other research[18,21], that considering together social isolation and loneliness will help represent better the social situation of older adults. This framework also provides a basis for exploring how distinct social experiences may relate to cognitive outcomes.

With regard to cognition, longitudinal studies and meta-analyses consistently demonstrate that social isolation and loneliness are associated with greater cognitive decline[22–24], highlighting the importance of considering these factors in cognitive aging research. Similarly, hearing impairment— both objective and subjective —has been extensively studied regarding its relation to cognitive functioning[3,4,8–10], with evidence suggesting that sensory decline relates to poorer cognitive outcomes and accelerated decline[25]. Given these findings, one may hypothesize that the combined presence of hearing impairment, social isolation and loneliness may be associated with cumulative and potentially synergistic patterns of cognitive vulnerability. Supporting this notion, prior research indicates that lonely older adults perceive everyday stressors as more distressing than their non-lonely peers[26], particularly when the stressor involves hearing loss[27]. Accordingly, investigating whether the strength of the association between hearing impairment and cognitive performance varies across distinct social isolation and loneliness profiles may enhance our understanding of individual differences in cognitive aging—both globally and across specific cognitive domains.

Despite the relevance of these factors, research exploring their combined associations with cognitive outcomes remains limited. One cross-sectional study examined whether social isolation and loneliness moderated the association between dual sensory impairment (visual and auditory) and cognition, but found no evidence supporting their role as moderators[28]. While some studies have documented associations among hearing impairment, social isolation, loneliness and cognition[10,23,28], relatively few studies have examined whether these social factors statistically moderate the relationship between hearing impairment and cognitive decline—that is, whether they are associated with variation in the strength of this association. Moreover, none of these studies have followed the framework proposed by Menec et al.[19,20], which emphasizes that social isolation and loneliness may represent distinct psychosocial experiences that are associated with health outcomes and should be examined together as profiles, rather than as separate predictors. Most existing research has treated social isolation and loneliness as independent variables, without investigating how together they may reflect distinct psychosocial experiences that can be associated with variation in the relationship between hearing impairment and cognitive decline. To address this gap, the present study adopts a person-centred approach using conceptually proposed and empirically validated profiles—non-isolated and not lonely, non-isolated and lonely, isolated and not lonely, and both isolated and lonely—to assess how the combination of these social factors, reflecting older adults' real-world social circumstances, is associated with differences in the strength of the relationship between hearing impairment and cognitive performance. A more nuanced investigation of these profiles may help identify individuals in later life who are at elevated risk of accelerated or more severe cognitive decline in the context of worsening hearing impairment.

Most prior studies attempting to disentangle these relationships have focused almost exclusively on episodic memory[23], leaving open the question of whether similar patterns extend to other cognitive domains supported by distinct neural systems. Executive functions, in particular, have received relatively little attention, despite their critical role in supporting independence and adaptive functioning in later life[29–31]. Given that age-related cognitive decline often affects executive processes such as cognitive flexibility and processing speed, the present study broadens the scope by examining both episodic memory and executive function. By leveraging longitudinal data, this study investigates whether distinct social isolation/loneliness profiles are associated with differences in the trajectory of cognitive functioning over time in the context of hearing impairment. Specifically, we address two key research questions: 1) Does the association between hearing impairment and cognition vary across different social isolation/loneliness profiles? 2) Are any observed differences between the four profiles consistent across cognitive domains (i.e., episodic memory, executive functions)? Drawing on 18 years of longitudinal data and accounting for the heterogeneity of the aging process[32,33], the study examines both between-person differences (i.e., average levels of hearing impairment and cognition across individuals) and within-person change (i.e., whether changes in hearing impairment over time are associated with concurrent changes in cognition). This dual-level approach allows us to investigate whether fluctuations in hearing impairment are related to cognitive trajectories, and whether these associations differ across social isolation/loneliness profiles. Understanding these complex interrelationships is critical for identifying individuals who may be at heightened risk of cognitive difficulties in later life, and for developing socially informed strategies to promote cognitive health in aging populations.

## Methods
### Respondents and sample
Data were derived from the Survey of Health, Ageing and Retirement in Europe (SHARE)[34]. SHARE biannually collects panel data of individuals residing in Europe and in Israel with a minimum age of 50 years. The first data collection occurred in 2004-2005 with the latest collection wave in 2021-2022 (study wave 9). SHARE collected retrospective life course data with the SHARE Life module in waves 3 and 7. Our study used data from wave 1 to wave 9, except for wave 3 in which no cognitive tests were performed.

**Table 1 | Descriptive statistics for study variables at first measurement (N = 33,741)**

| | M or % (SD or N) | Intraclass correlation coefficient (ICC) |
|---|---|---|
| Age | 61.4 (8.6) | 0.74 |
| Sex | | – |
| Male respondents | 42.9 (14,486) | |
| Female respondents | 57.1 (19,255) | |
| Education | 3.0 (1.4) | – |
| Chronic conditions | 1.5 (1.4) | 0.56 |
| Hearing impairment | 2.4 (1.0) | 0.47 |
| Social isolation × Loneliness profiles | | – |
| Non-isolated and low loneliness | 56.0 (18,891) | |
| Non-isolated and high loneliness | 41.5 (13,987) | |
| Isolated and low loneliness | 1.0 (346) | |
| Isolated and high loneliness | 1.5 (517) | |
| Immediate recall | 5.5 (1.7) | 0.48 |
| Delayed recall | 4.1 (2.0) | 0.53 |
| Verbal fluency | 20.5 (7.7) | 0.62 |

Respondents (initial sample N = 86,676) resided in one of the 12 following countries: Austria, Belgium, Czech Republic, Denmark, France, Germany, Greece, Italy, Poland, Spain, Sweden, and Switzerland. We selected these countries because they were the ones that participated consistently in the survey from the start and provided the most complete longitudinal data to investigate cognitive decline. The respondents were 50 years and older, with a mean age of M = 61.4 (SD = 8.6), a median age of 60, and an interquartile range of 54–67 (1st quartile = 54, 3rd quartile = 67). The age range spanned from 50 to 99 years at baseline. As we focused on uncorrected hearing impairment, respondents who reported that they had a hearing aid at any point during the study were excluded (N = 5,311). The final sample consisted of 33,741 respondents who had at least one observation in any of the variables used in the analysis (see Table 1 for descriptives on study variables). No ethnicity or race is assessed in SHARE.

All respondents provided a written informed consent before participating to the study and they were not financially remunerated, as they participate on a voluntary basis. SHARE was approved by the Ethics Council of the Max Planck Society and Ethics Research Committees in all participating countries. The study was not pre-registered.

## Measures

All measures used in this study were assessed by interviewers with the use of CAPI (Computer-Assisted Personal Interviewing), ensuring standardization and reducing potential biases related to literacy levels of respondents in different regions and cultures of Europe.

Outcomes: Three outcomes of cognitive functioning were assessed in every study wave and analysed: immediate recall, delayed recall[35] and verbal fluency[36]. Immediate recall assessed the extent to which the respondents were able to recall 10 words immediately after they were read out by the interviewer, within 1 minute. For delayed recall, respondents were again asked to recall the same 10 words within a 1-minute timeframe, after having completed other parts of the interview (approximately 10 minutes later). For both recall tasks, a higher number of recalled words indicated better cognitive functioning. For verbal fluency, respondents were asked to name as many animals as possible within 60 seconds. The same version of the question was asked in every wave, and the total score counted all animals named within the 60-seconds timeframe, regardless of whether they were real or mythical, a species or a specific breed of that species, the male or

female or infant name. A higher number of animals indicated better verbal fluency.

Predictors: Control variables assessed demographic information, including chronological age, sex (0 = male respondents, 1 = female respondents; inferred directly by the interviewer, the respondents did not report it on their own), and education, as well as chronic conditions. Education was assessed with the 1997 version of the International Standard Classification of Education[37], with 1 = primary education or first stage of basic education to 6 = second stage of tertiary education (e.g., PhD).

Based on previous research[38], respondents who undertake the same cognitive tests repeatedly across the study waves, tend to improve their performance, leading to potential underestimation of the effect of aging. Therefore, we estimated also retest effects using retest markers[38]. Specifically, we calculated binary indicators (0 = no, 1 = yes) of whether the task had been administered before, for each wave, excluding the first one.

Chronic conditions were assessed at each wave (from wave 1 on) in the physical health module with the following question: "Some people suffer from chronic or long-term health problems. By chronic or long-term we mean it has troubled you over a period of time or is likely to affect you over a period of time. Do you have any of such health problems, illness, disability or infirmity?". If an individual reported a new condition, this information was added to their existing number of chronic conditions.

Subjective hearing impairment was assessed in every wave (from wave 1 on). Respondents had to indicate how their hearing was on a 5-point Likert scale, ranging from 1 = excellent to 5 = poor. Higher scores indicated more severe subjective hearing impairment.

Social Isolation was assessed with the social connectedness scale[39,40] in wave 6. This was the first wave in which both social isolation and loneliness were concurrently measured, allowing for the creation of the profiles. The social connectedness scale is a summary scale incorporating five social network characteristics into one composite score. The characteristics include: 1) network size (representing the number of individuals that the respondent cited), 2) proximity (representing the number of cited individuals living within a 25 km range from the respondent), 3) frequency of contact (representing the number of cited individuals with a least weekly contact to the respondent), 4) support (representing the number of cited individuals with very or extremely close emotional ties), and, lastly, 5) diversity (representing the number of different types of relationships within the network). Network size, proximity, frequency of contact and support were assessed from 0 to 4 with 0 = 0 persons cited and 4 = 6 – 7 persons cited. Diversity also ranged from 0 to 4, with 4 representing the highest diversity in the network comprising of all 4 possible categories (i.e., spouse, other family including children, friend, and other). The final scale, provided by the SHARE study (Cronbach's alpha 0.92[39]), ranged from 0 = no social connectedness (0 social network members) to 4 = high social connectedness. Although social isolation can vary in degree and is sometimes treated as a continuous construct, it is often operationalized categorically in prior research—for example, by distinguishing between living alone versus living with others. Following this precedent and given our interest in identifying individuals at the most extreme end of the isolation spectrum, we created a binary indicator based on the social connectedness scale: individuals scoring 0 (indicating no social connections) were coded as 1 (isolated), and those scoring 1 or higher were coded as 0 (not isolated).

Loneliness was assessed with the 3-item UCLA scale[41] at wave 6. The mean composite score ranged from 3 = not lonely to 9 = very lonely (Cronbach's alpha = 0.76). Higher values indicated higher loneliness. A binary variable was computed based on the median value of the UCLA score with 0 = not lonely (scored less than 4 in the UCLA) and 1 = lonely (scored equal or more than 4 in the UCLA), as the continuous variable was highly skewed on the left and the median split ensured a balanced sample.

Social Isolation x Loneliness profiles. Inspired by past research[18,19,21], we created four static profiles based on the two binary indicators of social isolation and loneliness: 1) non-isolated and not lonely, 2) non-isolated and lonely, 3) isolated and not lonely, 4) isolated and lonely.

## Analytical strategy

We tested between-subjects' differences and within-subjects' change in immediate and delayed recall, as well as in verbal fluency, using multi-level modelling. To facilitate the interpretability of the results regarding the within subjects' change, we person-centred the time-varying variables (i.e., age, chronic conditions and hearing impairment). To investigate between-subjects differences of the time-varying variables we also calculated and included the across-waves person-mean in the models[42]. Sex and retest markers of cognitive outcomes were included in the model as time-invariant non-centred factors. Regarding hearing impairment, we were interested in the extent to which the linear and the quadratic change may relate to cognitive outcomes and therefore included a linear and a quadratic term.

We present the final and most parsimonious models (i.e., the ones with the best fit) which tested fixed and random effects, as well as interaction terms (see below). Estimates are reported unstandardized[42], while the effectsize package[43] was used to provide standardized estimates (in parentheses) with the "refit" method, refitting the model on z-scored variables. For the fixed effects, we reported 95% Confidence Intervals (CI) and $p$ values. For the random effects we reported Standard Deviations (SD) and their corresponding 95% Confidence Intervals (CI) for the variances and 95% Confidence Intervals (CI) for the covariances[44]. The fit of the models was tested with Akaike's Information Criterion (AIC), Bayesian Information Criterion (BIC) and -2 log likelihood (-2LL) fit indices (see Supplementary Tables 1–3 in the supplementary material). The Marginal $R^2$ was used to assess the variance explained only by the fixed effects and the Conditional $R^2$ was used to assess the variance explained by both the fixed and the random effects in each model. We included the different parameters in the model in a stepwise procedure: First, to verify the extent to which between-subjects differences and within-subjects change were related to the hierarchical clustering of the data, we tested a fully unconditional model, with no predictors included. Then, we added the fixed effects for all variables in the model, continuing with the inclusion of the random effects. The random effects included a random intercept and slope for the linear change and quadratic change of hearing impairment, allowing for individuals to have different baseline levels as well as varying patterns of within-person associations between hearing impairment and cognitive outcomes over time. Finally, we tested the three interaction terms of interest one by one, namely the mean, change and squared change parameters of hearing impairment with the profiles of loneliness/social isolation. The final models presented have retained only the interaction effects that improved the model fit. The equations for the final models were:

$Immediate\ Recall_{i,j} = \beta_0 + \beta_{1...7} \cdot retest_i + \beta_8 \cdot age_{devi,j} + \beta_9 \cdot age_{meani} + \beta_{10} \cdot sex_i + \beta_{11} \cdot education_i + \beta_{12} \cdot chronic\ conditions_{meani} + \beta_{13} \cdot chronic\ conditions_{devi,j} + \beta_{14} \cdot hearing\ impairment_{meani} + \beta_{15} \cdot hearing\ impairment_{devi,j} + \beta_{16} \cdot hearing\ impairment_{dev^2_{i,j}} + \beta_{17} \cdot profiles\ of\ isolation\ and\ loneliness_i + \beta_{18} \cdot (hearing\ impairment_{devi,j} \times profiles\ of\ isolation\ and\ loneliness_i) + u_{0i} + u_{1i} \cdot hearing\ impairment_{devi,j} + u_{2i} \cdot hearing\ impairment_{dev^2_{i,j}} + \epsilon_{i,j}$

$Delayed\ Recall_{i,j} = \beta_0 + \beta_{1...7} \cdot retest_i + \beta_8 \cdot age_{dev\ i,j} + \beta_9 \cdot age_{mean\ i} + \beta_{10} \cdot sex_i + \beta_{11} \cdot education_i + \beta_{12} \cdot chronic\ conditions_{mean\ i} + \beta_{13} \cdot chronic\ conditions_{dev\ i,j} + \beta_{14} \cdot hearing\ impairment_{mean\ i} + \beta_{15} \cdot hearing\ impairment_{dev\ i,j} + \beta_{16} \cdot hearing\ impairment_{dev^2_{i,j}} + \beta_{17} \cdot profiles\ of\ isolation\ and\ loneliness_i + \beta_{18} \cdot (hearing\ impairment_{dev\ i,j} \times profiles\ of\ isolation\ and\ loneliness_i) + u_{0i} + u_{1i} \cdot hearing\ impairment_{dev\ i,j} + u_{2i} \cdot hearing\ impairment_{dev^2_{i,j}} + \epsilon_{i,j}$

$Verbal\ Fluency_{i,j} = \beta_0 + \beta_{1...7} \cdot retest_i + \beta_8 \cdot age_{dev\ i,j} + \beta_9 \cdot age_{mean\ i} + \beta_{10} \cdot sex_i + \beta_{11} \cdot education_i + \beta_{12} \cdot chronic\ conditions_{mean\ i} + \beta_{13} \cdot chronic\ conditions_{dev\ i,j} + \beta_{14} \cdot hearing\ impairment_{mean\ i} + \beta_{15} \cdot hearing\ impairment_{dev\ i,j} + \beta_{16} \cdot hearing\ impairment_{dev^2_{i,j}} + \beta_{17} \cdot profiles\ of\ isolation\ and\ loneliness_i + u_{0i} + u_{1i} \cdot hearing\ impairment_{dev\ i,j} + u_{2i} \cdot hearing\ impairment_{dev^2_{i,j}} + \epsilon_{i,j}$

In equations 1, 2, and 3, $\beta_0$ represents the intercept, while $\beta_1$, $\beta_2$, …, $\beta_{17}$ are the fixed effects coefficients for each predictor variable. $\beta_{18}$ denotes the coefficient for the interaction term between hearing impairment (deviation) and profiles of social isolation and loneliness, capturing how the effect of hearing impairment (deviation) on the outcome changes depending on the level of the profiles of social isolation and loneliness. $u_{01}$ is the random intercept for each individual (indexed by i), accounting for individual differences in baseline scores. The terms $u_{1i}$ and $u_{2i}$ represent random slopes for each individual, associated with hearing impairment (deviation) and hearing impairment (deviation$^2$), allowing for individual-specific trajectories in the effect of these variables. We also allow random effects to correlate. Finally, $\epsilon_{i,j}$ is the residual error term for each observation j within individual i, capturing any remaining unexplained variability.

To address our second research question—whether the potential differences across social isolation and loneliness profiles on the link between change in hearing impairment and cognition differs across cognitive domains—we conducted a multivariate linear multilevel modelling analysis using cognitive scores as the outcome. The cognitive scores were z-standardized to facilitate comparison across domains as the recall tests differed in their answering format from that of verbal fluency. The model included a categorical variable indicating cognitive domain (immediate recall, delayed recall, and verbal fluency), with immediate recall set as the reference category. This model incorporated all two-way and three-way interactions between cognitive domain, loneliness/social isolation profiles, and time-varying hearing impairment (linear change). Fixed effects also included within- and between-subjects' age, gender, education, number of chronic conditions (between- and within-subjects' terms), hearing impairment (between- and within-person linear and quadratic terms), and practice effects using binary retest indicators for waves 2–9. The model included random intercepts and slopes (hearing impairment between- and within-subjects' linear and quadratic terms). This analytic strategy allowed us to test whether the associations between psychosocial vulnerability, hearing impairment, and cognitive functioning differed significantly across memory and executive function domains. The data met the assumptions of the statistical tests used (e.g., normality tested). All models were computed using the Restricted Maximum Likelihood estimation method. All analyses were conducted with R[45], and the lme4 package[44].

## Reporting summary

Further information on research design is available in the Nature Portfolio Reporting Summary linked to this article.

## Results

Descriptive statistics are presented in Table 1 (for correlations between study variables at the first measurement point see Supplementary Table 4 in the supplementary material). The three fully unconditional models (no predictors) revealed that the ICC (Intraclass Correlation Coefficient) for immediate recall was 0.48, 0.53 for delayed recall, and 0.62 for verbal fluency, indicating that 48%, 53%, and 62% of the cognitive outcomes varied across individuals, hence justifying the use of multilevel models. We therefore proceeded to more complex models to test between-subjects differences and within-subjects change. The most parsimonious model for each cognitive outcome is presented in Tables 2–4.

### Immediate recall

We first present results regarding the test-retest effects, then the between subjects' differences, the within subjects' change, the interactions, and finally the random effects (Table 2). Test-retest effects revealed a significant improvement in immediate recall scores over repeated assessments, indicating practice effects across waves. Specifically, compared to baseline, respondents demonstrated a steady increase in performance across multiple testing sessions, with scores improving by 0.07 points in Wave 2 ($b = 0.07$, 95% CI [0.03, 0.10]), reaching 0.23 points higher at Wave 5 ($b = 0.23$, 95% CI [0.20, 0.26]). Regarding between subjects' differences, individuals who were

**Table 2 | Multilevel model with fixed and random effects for immediate recall**

| | Immediate recall B (β) | SE | t (df) | CI | p |
|---|---|---|---|---|---|
| **Fixed effects** | | | | | |
| (Intercept) | 7.62 (0.04) | 0.06 | 127.71 (35,630) | 7.50 to 7.74 | <0.001 |
| Retest effects | | | | | |
| Wave 2 | 0.07 (0.01) | 0.02 | 3.85 (112,600) | 0.03 to 0.10 | <0.001 |
| Wave 4 | 0.22 (0.03) | 0.02 | 13.52 (115,500) | 0.19 to 0.25 | <0.001 |
| Wave 5 | 0.23 (0.05) | 0.01 | 16.83 (119,900) | 0.20 to 0.26 | <0.001 |
| Wave 6 | 0.21 (0.05) | 0.01 | 14.98 (125,300) | 0.18 to 0.24 | <0.001 |
| Wave 7 | 0.22 (0.03) | 0.02 | 9.98 (130,000) | 0.18 to 0.26 | <0.001 |
| Wave 8 | 0.18 (0.03) | 0.02 | 8.17 (135,900) | 0.13 to 0.22 | <0.001 |
| Wave 9 | 0.20 (0.04) | 0.02 | 8.04 (136,300) | 0.15 to 0.25 | <0.001 |
| Between-subjects' effects | | | | | |
| Age (M) | −0.05 (−0.23) | 0.001 | −62.17 (34,410) | −0.05 to −0.05 | <0.001 |
| Sex (Female respondents) | 0.31 (0.09) | 0.01 | 24.88 (32,340) | 0.28 to 0.33 | <0.001 |
| Education | 0.33 (0.27) | 0.004 | 74.15 (32,300) | 0.32 to 0.33 | <0.001 |
| Chronic conditions (M) | −0.05 (−0.04) | 0.01 | −10.26 (33,530) | −0.06 to −0.04 | <0.001 |
| Hearing impairment (M) | −0.16 (−0.07) | 0.01 | −19.48 (33,250) | −0.18 to −0.15 | <0.001 |
| Profiles of social isolation and loneliness (ref: non-isolated and low loneliness) | | | | | |
| Non-isolated and high loneliness | −0.18 (−0.10) | 0.01 | −13.96 (32,370) | −0.20 to −0.15 | <0.001 |
| Isolated and low loneliness | −0.23 (−0.02) | 0.06 | −3.82 (33,900) | −0.35 to −0.11 | <0.001 |
| Isolated and high loneliness | −0.38 (−0.22) | 0.05 | −7.63 (33,560) | −0.48 to −0.28 | <0.001 |
| Within-subjects' effects | | | | | |
| Age (C) | −0.04 (−0.09) | 0.002 | −18.18 (135,500) | −0.04 to −0.03 | <0.001 |
| Chronic conditions (C) | −0.01 (−0.01) | 0.004 | −2.38 (103,700) | −0.02 to −0.001 | 0.017 |
| Hearing impairment (C) | −0.03 (−0.01) | 0.01 | −4.15 (17,050) | −0.05 to −0.02 | <0.001 |
| Hearing impairment ($C^2$) | −0.06 (−0.02) | 0.01 | −8.35 (4805) | −0.07 to −0.04 | <0.001 |
| Interaction effects | | | | | |
| Hearing impairment (C) * Non-isolated and high loneliness | −0.06 (−0.02) | 0.01 | −5.35 (16,170) | −0.09 to −0.04 | <0.001 |
| Hearing impairment (C) * Isolated and low loneliness | 0.01 (0.01) | 0.06 | 0.22 (16,430) | −0.10 to 0.13 | 0.823 |
| Hearing impairment (C) * Isolated and high loneliness | −0.04 (−0.01) | 0.05 | −0.74 (16,190) | −0.13 to 0.06 | 0.462 |
| | **Estimates** | **SD** | **CI** | | |
| **Random effects** | | | | | |
| Residual variance | 1.45 | 1.21 | 1.20 to 1.21 | | |
| Intercept (variance) | 0.82 | 0.91 | 0.89 to 0.92 | | |
| Hearing Impairment (C) slope (variance) | 0.08 | 0.28 | 0.25 to 0.31 | | |
| Hearing Impairment ($C^2$) slope (variance) | 0.01 | 0.12 | 0.06 to 0.16 | | |
| Intercept*Hearing impairment (C) slope (covariance) | 0.11 | – | 0.06 to 0.16 | | |
| Intercept*Hearing impairment ($C^2$) slope (covariance) | −0.25 | – | −0.41 to −0.13 | | |
| Hearing impairment (C) slope* Hearing impairment ($C^2$) slope (covariance) | −0.13 | – | −0.40 to 0.11 | | |
| ICC | 0.37 | | | | |
| N | 33,726 | | | | |
| Observations | 137,005 | | | | |
| Marginal $R^2$/Conditional $R^2$ | 0.209/0.498 | | | | |

The CIs in the random effects' variances correspond to their Standard Deviations, while in the random effects' covariances they correspond to the actual estimates.

*M* person-mean variable (between-subjects differences), *C* person-mean centred variable indicating the linear change, $C^2$ person-mean centred variable indicating the quadratic change, *SD* standard deviation, *CI* 95% confidence intervals, *Marginal* $R^2$ variance explained by fixed effects only, *Conditional* $R^2$ variance explained by fixed and random effects.

younger in age (*b* = −0.05, 95% CI [−0.05, −0.05]), of female sex (*b* = 0.31, 95% CI [0.28, 0.33]), highly educated (*b* = 0.33, 95% CI [0.32, 0.33]), and with fewer chronic conditions (*b* = −0.05, 95% CI [−0.06, −0.04]) scored better in the immediate recall task. Additionally, individuals with greater hearing impairment exhibited 0.16-point lower immediate recall performance than those with better hearing ability (*b* = −0.16, 95% CI [−0.18, −0.15]). Regarding social isolation and loneliness, individuals belonging to any of the three disadvantaged profiles (i.e., non-isolated but lonely, isolated but not lonely, isolated and lonely) exhibited poorer immediate recall performance compared to non-isolated and not lonely

**Fig. 1 | Immediate recall as a function of within-person change in hearing impairment, by profiles of social isolation and loneliness.** Predicted values are plotted with standard errors. Hearing impairment is centered at each individual's mean. $N = 33,726$ respondents.

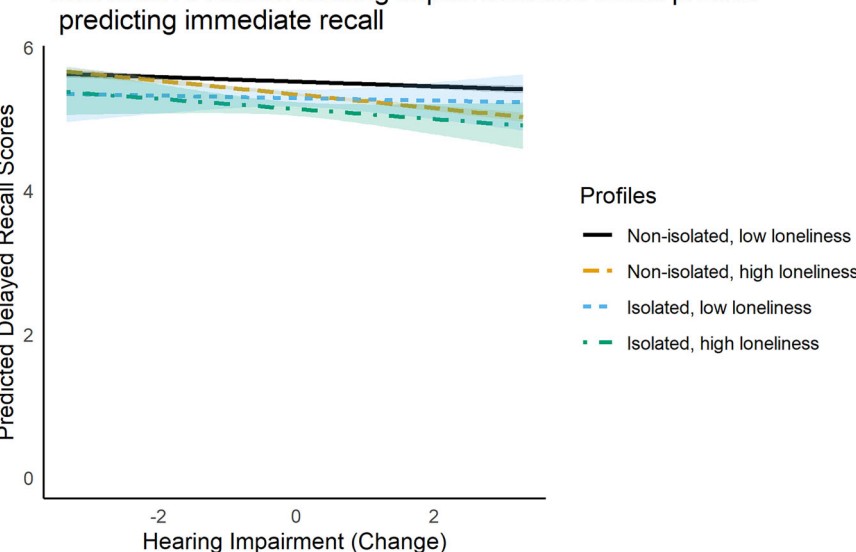

individuals. The largest effect was observed among individuals who were both socially isolated and lonely. This profile had the lowest average performance, with scores that were 0.38 points lower than those in the reference group ($b = -0.38$, 95% CI [$-0.48$, $-0.28$]). The next largest differences were observed among those who were isolated but not lonely ($b = -0.23$, 95% CI [$-0.35$, $-0.11$]) and those who were non-isolated but lonely ($b = -0.18$, 95% CI [$-0.20$, $-0.15$]).

The within-subjects' effects (time varying person-centred predictors) showed that each additional year of age was associated with a 0.04-point decrease in immediate recall ($b = -0.04$, 95% CI [$-0.04$, $-0.03$]). Over a 10-year period, this would correspond to a decline of 0.40 points, suggesting that, on average, an individual in their 70s would recall almost one less word than they did in their 60s in the immediate recall task. Similarly, each additional chronic condition was linked to a 0.01-point decline in immediate recall ($b = -0.01$, 95% CI [$-0.02$, $-0.001$]). For an individual experiencing an increase of three chronic conditions (e.g., developing diabetes, hypertension, and cardiovascular disease), this would correspond to a 0.03-point decrease, which, while modest, may contribute to compounded cognitive burden over time, particularly in individuals already at risk due to age or other factors. Hearing impairment had both a linear and quadratic effect on immediate recall (see Supplementary Fig. 1 in the supplementary material). Each one-unit increase in hearing impairment was associated with a 0.03-point decrease in immediate recall ($b = -0.03$, 95% CI [$-0.05$, $-0.02$]). For example, an individual experiencing a 4-unit decline (e.g., from excellent to poor hearing) would be expected to recall 0.12 fewer words, on average. Furthermore, the quadratic term was also significant ($b = -0.06$, 95% CI [$-0.07$, $-0.04$]), suggesting that more severe hearing impairment is associated with an accelerated rate of decline in recall ability. In other words, while individuals with mild hearing loss may experience gradual cognitive decline, those with more severe hearing impairment face an increasingly steeper drop in recall ability.

Regarding the interaction effects, testing the interplay between the time varying hearing impairment with the profiles of social isolation and loneliness (non-time-varying predictor), individuals in the non-isolated but lonely profile experienced worse decline in recall scores as hearing impairment increased, as shown also in Fig. 1. For individuals in this profile, each unit increase in hearing impairment was associated with an additional 0.06-point decline in recall scores ($b = -0.06$, 95% CI [$-0.09$, $-0.04$]), meaning that an individual who progresses from excellent to poor hearing (4-unit decline) would recall approximately 0.24 fewer words. This decline does not occur in isolation. It adds to the declines already associated with aging (0.04 words lost per year of age) and hearing impairment itself (0.03

words lost per unit increase in hearing impairment, accelerating further due to quadratic effects). Therefore, as an example, an average individual with a mean score at baseline of 5.5 (range 1-10 in delayed recall) in 10 years may experience a 0.40-point decline related to aging alone and a further decline of 0.12-points due to a potential 4 unit increase in hearing impairment, scoring approximately 4.98, even without considering the accelerated decline. For the lonely-in-the-crowd profile with a similar increase in hearing impairment, an additional 0.24-point decline in delayed recall scores can be added, resulting in a cognitive score of 4.74 which would be approximately 14% lower than its baseline.

Regarding the random effects, the within-subjects' random variance was significant ($\sigma^2 = 1.45$, SD = 1.21, 95% CI [1.20, 1.21]), indicating that there was significant variability in the immediate recall scores within individuals over time. Similarly, the random intercept variance was significant ($\sigma^2 = 0.82$, SD = 0.91, 95% CI [0.89, 0.92]), suggesting that there was significant variability between individuals with regards to their initial level of immediate recall scores. The random slopes of the linear and quadratic change of hearing impairment also varied significantly, indicating that the rate of change in the linear and the quadratic effects differed between individuals. Specifically, the variance of the linear slope of hearing impairment was 0.08 (SD = 0.28; 95% CI [0.25, 0.31]), and the variance of the quadratic slope was 0.01 (SD = 0.12; 95% CI [0.06, 0.16]), confirming that the degree of cognitive decline associated with hearing impairment was not uniform across respondents. Moreover, the covariances between the intercept and the slopes of linear and quadratic change in hearing revealed two important patterns: 1) For individuals with higher baseline levels of immediate recall, their cognitive performance declined less steeply as hearing impairment worsened, compared to those with lower baseline recall scores, who exhibited a more pronounced decline (*intercept-hearing impairment linear slope covariance* = 0.11, 95% CI [0.06, 0.16]). 2) Similarly, individuals with higher baseline immediate recall scores showed a less pronounced curvature in the hearing impairment-recall relationship, whereas those with lower baseline recall exhibited a steeper non-linear decline (*intercept-hearing impairment quadratic slope covariance* = $-0.25$, 95% CI [$-0.41$, $-0.13$]).

**Delayed recall**

In delayed recall (Table 3), the test-retest effects revealed a significant improvement across waves, ranging from 0.16 (Wave 2) to 0.36 (Wave 4) points of increase compared to the baseline score of delayed recall. The fixed main effects findings were almost identical to the immediate recall results for both between subjects' differences and within-subjects' change. Individuals

**Table 3 | Multilevel model with fixed and random effects for delayed recall**

| | Delayed recall B (β) | SE | t (df) | CI | p |
|---|---|---|---|---|---|
| **Fixed effects** | | | | | |
| (Intercept) | 6.65 (0.04) | 0.08 | 87.03 (35,770) | 6.52 to 6.82 | <0.001 |
| Retest effects | | | | | |
| Wave 2 | 0.16 (0.02) | 0.02 | 7.50 (111,000) | 0.12 to 0.20 | <0.001 |
| Wave 4 | 0.36 (0.04) | 0.02 | 19.09 (114,300) | 0.33 to 0.40 | <0.001 |
| Wave 5 | 0.34 (0.05) | 0.02 | 20.54 (118,800) | 0.30 to 0.37 | <0.001 |
| Wave 6 | 0.35 (0.07) | 0.02 | 20.88 (124,000) | 0.32 to 0.38 | <0.001 |
| Wave 7 | 0.29 (0.03) | 0.03 | 10.95 (129,200) | 0.23 to 0.34 | <0.001 |
| Wave 8 | 0.26 (0.04) | 0.03 | 10.00 (135,000) | 0.21 to 0.31 | <0.001 |
| Wave 9 | 0.26 (0.04) | 0.03 | 8.86 (135,700) | 0.21 to 0.32 | <0.001 |
| Between-subjects' effects | | | | | |
| Age (M) | −0.06 (−0.23) | 0.001 | −60.68 (34,700) | −0.06 to −0.06 | <0.001 |
| Sex (Female respondents) | 0.44 (0.10) | 0.02 | 27.75 (32,900) | 0.41 to 0.47 | <0.001 |
| Education | 0.39 (0.27) | 0.01 | 69.58 (32,890) | 0.38 to 0.41 | <0.001 |
| Chronic conditions (M) | −0.07 (−0.04) | 0.01 | −11.22 (33,950) | −0.09 to −0.06 | <0.001 |
| Hearing impairment (M) | −0.17 (−0.06) | 0.01 | −15.74 (33,860) | −0.19 to −0.15 | <0.001 |
| Profiles of social isolation and loneliness (ref: non-isolated and low loneliness) | | | | | |
| Non-isolated and high loneliness | −0.23 (−0.11) | 0.02 | −14.12 (32,940) | −0.26 to −0.20 | <0.001 |
| Isolated and low loneliness | −0.24 (−0.11) | 0.08 | −3.07 (34,250) | −0.39 to −0.09 | 0.002 |
| Isolated and high loneliness | −0.45 (−0.21) | 0.06 | −6.97 (33,950) | −0.57 to −0.32 | <0.001 |
| Within-subjects' effects | | | | | |
| Age (C) | −0.04 (−0.07) | 0.002 | −15.92 (134,200) | −0.04 to −0.03 | <0.001 |
| Chronic conditions (C) | −0.01 (−0.01) | 0.01 | −2.52 (103,800) | −0.02 to −0.001 | 0.012 |
| Hearing impairment (C) | −0.05 (−0.01) | 0.01 | −5.19 (16,880) | −0.07 to −0.03 | <0.001 |
| Hearing impairment (C$^2$) | −0.06 (−0.02) | 0.01 | −7.39 (4863) | −0.07 to −0.04 | <0.001 |
| Interaction effects | | | | | |
| Hearing impairment (C) * Non-isolated and high loneliness | −0.04 (−0.01) | 0.01 | −3.03 (16,080) | −0.07 to −0.01 | 0.002 |
| Hearing impairment (C) * Isolated and low loneliness | 0.02 (0.01) | 0.07 | 0.26 (16,300) | −0.12 to 0.15 | 0.797 |
| Hearing impairment (C) * Isolated and high loneliness | −0.08 (−0.02) | 0.06 | −1.42 (15,810) | −0.19 to 0.03 | 0.156 |
| | Estimates | SD | | CI | |
| **Random effects** | | | | | |
| Residual variance | 2.01 (1.42) | 1.42 | | 1.20 to 2.43 | |
| Intercept (variance) | 1.48 (1.22) | 1.22 | | 1.20 to 1.23 | |
| Hearing Impairment (C) slope (variance) | 0.12 (0.34) | 0.34 | | 0.31 to 0.37 | |
| Hearing Impairment (C$^2$) slope (variance) | 0.02 (0.14) | 0.14 | | 0.09 to 0.19 | |
| Intercept*Hearing impairment (C) slope (covariance) | 0.13 | – | | 0.08 to 0.19 | |
| Intercept*Hearing impairment (C$^2$) slope (covariance) | −0.34 | – | | −0.68 to −0.09 | |
| Hearing impairment (C) slope* Hearing impairment (C$^2$) slope (covariance) | −0.32 | – | | −0.51 to −0.22 | |
| ICC | 0.43 | | | | |
| N | 33,725 | | | | |
| Observations | 137,039 | | | | |
| Marginal R$^2$/Conditional R$^2$ | 0.207/0.545 | | | | |

The CIs in the random effects' variances correspond to their Standard Deviations, while in the random effects' covariances they correspond to the actual estimates.

M person-mean variable (between-subjects differences), C person-mean centred variable indicating the linear change, C$^2$ person-mean centred variable indicating the quadratic change, SD standard deviation, CI 95% confidence intervals, Marginal R$^2$ variance explained by fixed effects only, Conditional R$^2$ variance explained by fixed and random effects.

who were younger ($b = -0.06$, 95% CI [$-0.06$, $-0.06$]), of female sex ($b = 0.44$, 95% CI [0.41, 0.47]), and more highly educated ($b = 0.39$, 95% CI [0.38, 0.41]) exhibited better delayed recall performance. Additionally, individuals with fewer chronic conditions ($b = -0.07$, 95% CI [$-0.09$, $-0.06$]) and better hearing ability ($b = -0.17$, 95% CI [$-0.19$, $-0.15$]) had

significantly higher delayed recall scores. Regarding the social isolation and loneliness profiles, individuals in any of the three disadvantaged profiles performed worse than the non-isolated and not lonely reference group. Those in the non-isolated but lonely profile performed 0.23 points lower ($b = -0.23$, 95% CI [$-0.26$, $-0.20$]), the isolated but not lonely profile

**Fig. 2 | Delayed recall as a function of within-person change in hearing impairment, by profiles of social isolation and loneliness.** Predicted values are plotted with standard errors. Hearing impairment is centered at each individual's mean. N = 33,725 respondents.

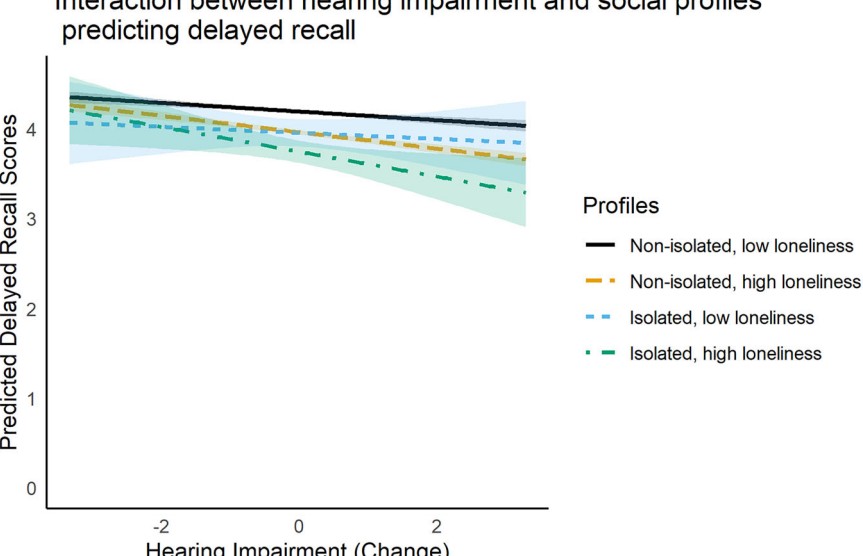

scored 0.24 points lower ($b = -0.24$, 95% CI [$-0.39$, $-0.09$]), and the most vulnerable group—isolated and lonely individuals—scored 0.45 points lower ($b = -0.45$, 95% CI [$-0.57$, $-0.32$]), suggesting that the co-occurrence of social isolation and loneliness is associated with lower cognitive performance.

Within subjects' effects showed that a decline in delayed recall performance was observed over time. Each additional year of aging was associated with a 0.08-point decline in recall performance ($b = -0.04$, 95% CI [$-0.04$, $-0.03$]) and each additional chronic condition resulted in a 0.01-point decrease ($b = -0.01$, 95% CI [$-0.02$, $-0.001$]). Increases in hearing impairment were associated with declines in delayed recall performance, with each unit of worsening hearing impairment linked to a 0.05-point reduction in delayed recall performance ($b = -0.05$, 95% CI [$-0.07$, $-0.03$]). Finally, the quadratic effect of hearing impairment revealed an accelerated decline ($b = -0.06$, 95% CI [$-0.07$, $-0.04$]) as hearing impairment worsens (see Supplementary Fig. 2 in the supplementary material).

Regarding the interactions, only the interaction between the linear effect of hearing impairment and the loneliness/social isolation profiles improved the model and was, therefore, maintained in the final one (Fig. 2). Similar to the immediate recall model, the lonely-in-the-crowd profile (non-isolated but lonely) showed a steeper decline in delayed recall scores with increasing hearing impairment compared to the non-isolated and not lonely reference profile. Specifically, each unit of increase in hearing impairment was associated with an additional 0.04-point decline in delayed recall performance for this profile ($b = -0.04$, 95% CI [$-0.07$, $-0.01$]). Thus, an individual who progresses from excellent to poor hearing (4-unit decline) would experience a 0.16-point reduction in recall scores due to this interaction alone. This decline adds to that already associated with age (0.40 points per decade) and hearing impairment itself (0.05 points per unit increase in impairment, with further acceleration due to the quadratic effect), compounding the overall reduction in cognitive performance over time. As a result, an individual with a baseline delayed recall score of 4.1 (on a 1-10 scale) could expect a total reduction of approximately 0.40 points due to aging, 0.20 points due to hearing impairment, and an additional 0.16 points if they are in the lonely but non-isolated profile. For illustrative purposes, this combined decline translates to a 0.80-point loss, representing approximately 20% of their baseline cognitive score over a decade, without adding the quadratic effect or that of increasing chronic conditions.

Regarding the random effects, the within-subjects' random variance was significant ($\sigma^2 = 2.01$, SD = 1.42, 95% CI [1.20, 2.43]), indicating substantial variability in delayed recall scores within individuals over time.

Similarly, the random intercept variance was significant ($\sigma^2 = 1.48$, SD = 1.22, 95% CI [1.20, 1.23]), suggesting notable between-subject differences in initial delayed recall ability. The random slopes for the linear and quadratic effects of hearing impairment also varied significantly across individuals (*linear slope variance* = 0.12, SD = 0.34, 95% CI [0.31, 0.37]; *quadratic slope variance* = 0.02, SD = 0.14, 95% CI [0.09, 0.19]), indicating substantial individual differences in the rate of cognitive decline. Two patterns emerged from the covariances between baseline scores and hearing impairment slopes: (1) Individuals with higher baseline recall exhibited a less steep decline in cognitive scores as hearing impairment worsened, compared to those with lower initial recall, who experienced sharper declines (*intercept-hearing impairment linear slope covariance* = 0.06, 95% CI [0.08, 0.19]); and (2) individuals with higher initial recall showed a less pronounced acceleration in decline, while those with lower recall had a steeper non-linear decline (*intercept-hearing impairment quadratic slope covariance* = -0.06, 95% CI [$-0.68$, $-0.09$]). Finally, the negative covariance between the linear and quadratic slopes ($-0.02$, 95% CI [$-0.51$, $-0.22$]) suggests that individuals experiencing steeper initial declines in delayed recall due to hearing impairment were more likely to show less acceleration in their cognitive deterioration over time.

### Verbal fluency

Similar to immediate recall, test-retest effects revealed a measurable improvement across waves, ranging from 0.48 (Wave 4) to 1.98 (Wave 9) points of increase compared to the baseline score of verbal fluency (Table 4). Between subjects' fixed effects indicated that younger individuals ($b = -0.16$, 95% CI [$-0.17$, $-0.15$]), women ($b = 0.38$, 95% CI [0.26, 0.51]), and higher education ($b = 1.65$, 95% CI [1.60, 1.69]) were related to better verbal fluency scores. We found no statistically significant evidence regarding the link between the overall number of chronic conditions and the verbal fluency score ($b = -0.04$, 95% CI [$-0.10$, 0.01]). Additionally, individuals with better hearing ability had overall better scores in verbal fluency ($b = -0.49$, 95% CI [$-0.58$, $-0.41$]). Regarding the profiles of social isolation and loneliness, in this case, both the non-isolated but lonely individuals and those who were isolated and lonely exhibited lower fluency scores compared to the non-isolated and not lonely reference profile. Specifically, the non-isolated but lonely profile performed 1.49 points lower ($b = -1.49$, 95% CI [$-1.61$, $-1.36$]) and the most vulnerable group—isolated and lonely individuals—scored 1.94 points lower ($b = -1.94$, 95% CI [$-2.45$, $-1.43$]).

Regarding the within-subjects' effects, results showed that with each additional year of aging, individuals experienced a 0.20-word decrease in

**Table 4 | Multilevel model with fixed and random effects for verbal fluency**

| | Verbal fluency B (β) | SE | t (df) | CI | p |
|---|---|---|---|---|---|
| **Fixed effects** | | | | | |
| (Intercept) | 26.75 (0.06) | 0.31 | 86.76 (35,110) | 26.15 to 27.36 | <0.001 |
| Retest effects | | | | | |
| Wave 2 | 0.51 (0.01) | 0.07 | 7.24 (106,900) | 0.37 to 0.65 | <0.001 |
| Wave 4 | 0.48 (0.02) | 0.06 | 7.58 (111,100) | 0.36 to 0.61 | <0.001 |
| Wave 5 | 1.08 (0.05) | 0.05 | 19.72 (115,600) | 0.97 to 1.19 | <0.001 |
| Wave 6 | 1.24 (0.07) | 0.06 | 21.88 (120,200) | 1.13 to 1.35 | <0.001 |
| Wave 7 | 1.32 (0.04) | 0.09 | 14.85 (123,600) | 1.14 to 1.49 | <0.001 |
| Wave 8 | 1.85 (0.08) | 0.09 | 20.98 (130,600) | 1.68 to 2.03 | <0.001 |
| Wave 9 | 1.98 (0.08) | 0.10 | 19.30 (131,700) | 1.78 to 2.18 | <0.001 |
| Between-subjects' effects | | | | | |
| Age (M) | −0.16 (−0.17) | 0.004 | −40.31 (34,390) | −0.17 to −0.15 | <0.001 |
| Sex (Female respondents) | 0.38 (0.02) | 0.06 | 5.97 (33,100) | 0.26 to 0.51 | <0.001 |
| Education | 1.65 (0.30) | 0.02 | 71.59 (33,120) | 1.60 to 1.69 | <0.001 |
| Chronic conditions (M) | −0.04 (−0.01) | 0.03 | −1.65 (33,890) | −0.10 to 0.01 | 0.099 |
| Hearing impairment (M) | −0.49 (−0.05) | 0.04 | −11.37 (33,830) | −0.58 to −0.41 | <0.001 |
| Profiles of social isolation and loneliness (ref: non-isolated and low loneliness) | | | | | |
| Non-isolated and high loneliness | −1.49 (−0.19) | 0.07 | −22.61 (33,130) | −1.61 to −1.36 | <0.001 |
| Isolated and low loneliness | −0.23 (−0.03) | 0.32 | −0.72 (34,130) | −0.84 to 0.39 | 0.474 |
| Isolated and high loneliness | −1.94 (−0.25) | 0.26 | −7.50 (33,820) | −2.45 to −1.43 | <0.001 |
| Within-subjects' effects | | | | | |
| Age (C) | −0.20 (−0.11) | 0.01 | −25.30 (129,200) | −0.22 to −0.19 | <0.001 |
| Chronic conditions (C) | 0.04 (0.004) | 0.02 | 2.46 (103,500) | 0.01 to 0.07 | 0.014 |
| Hearing impairment (C) | −0.25 (−0.02) | 0.02 | −10.69 (16,740) | −0.29 to −0.20 | <0.001 |
| Hearing impairment (C$^2$) | −0.18 (−0.01) | 0.03 | −6.62 (5,018) | −0.23 to −0.13 | <0.001 |
| | **Estimates** | **SD** | | **CI** | |
| **Random effects** | | | | | |
| Residual variance | 22.07 | 4.70 | | 4.68 to 4.72 | |
| Intercept (variance) | 27.22 | 5.22 | | 5.16 to 5.27 | |
| Hearing Impairment (C) slope (variance) | 1.20 | 1.09 | | 0.99 to 1.19 | |
| Hearing Impairment (C$^2$) slope (variance) | 0.34 | 0.58 | | 0.42 to 0.72 | |
| Intercept*Hearing impairment (C) slope (covariance) | 0.04 | – | | −0.01 to 0.09 | |
| Intercept*Hearing impairment (C$^2$) slope (covariance) | −0.27 | – | | −0.37 to −0.18 | |
| Hearing impairment (C) slope* Hearing impairment (C$^2$) slope (covariance) | 0.24 | – | | 0.04 to 0.45 | |
| ICC | 0.55 | | | | |
| N | 33,727 | | | | |
| Observations | 137,031 | | | | |
| Marginal R$^2$/Conditional R$^2$ | 0.182/0.634 | | | | |

The CIs in the random effects' variances correspond to their Standard Deviations, while in the random effects' covariances they correspond to the actual estimates.

*M* person-mean variable (between-subjects differences), *C* person-mean centred variable indicating the linear change, *C$^2$* person-mean centred variable indicating the quadratic change, *SD* standard deviation, *CI* 95% confidence intervals, *Marginal R$^2$* variance explained by fixed effects only, *Conditional R$^2$* variance explained by fixed and random effects.

verbal fluency ($b = -0.20$, 95% CI [−0.22, −0.19]), indicating that individuals tend to experience declines in verbal fluency as they grow older. Over a 10-year period, this translates to a loss of 2.0 words on the verbal fluency task due to aging alone. The change in chronic conditions was associated to the decline in verbal fluency scores ($b = 0.04$, 95% CI [0.01, 0.07]). Hearing impairment had both a linear and quadratic effect on verbal fluency (see Supplementary Fig. 3 in the supplementary material), where each unit increase in hearing impairment was associated with a 0.25-word reduction in fluency ($b = -0.25$, 95% CI [−0.29, −0.20]), suggesting that a severe deterioration in hearing (e.g., going from excellent hearing = 1 to poor hearing = 5) would result in an associated loss of one word ($0.25 \times 4 = 1.00$). Moreover, the quadratic term ($b = -0.18$, 95% CI [−0.23, −0.13]) indicated that as hearing impairment worsened, the decline in verbal fluency accelerated rather than remained linear. This suggests that cognitive declines may accelerate disproportionately in individuals with more severe hearing impairment compared to those with only mild hearing impairment. None of the interaction terms between hearing impairment (both linear and quadratic, between- and within-subjects) and the social isolation/loneliness profiles improved model fit. These were therefore excluded from the final model.

Regarding the random effects for verbal fluency, the within subjects' variance ($\sigma^2 = 22.07$, SD = 4.70), 95% CI [1.42, 1.43]), indicating substantial variability in verbal fluency scores within individuals over time. The random intercept variance was also significant ($\sigma^2 = 27.22$, SD = 5.22, 95% CI [5.16, 5.27]), suggesting notable between-subject differences in initial verbal fluency performance. The random slopes for the linear and quadratic effects of hearing impairment also varied significantly across individuals (*linear slope variance* = 1.20, SD = 1.09, 95% CI [0.99, 1.19]; *quadratic slope variance* = 0.34, SD = 0.58, 95% CI [0.42, 0.72]), confirming that not all individuals experience cognitive decline at the same rate. The covariance between the intercept and the linear slope of hearing impairment was not significant (0.04, 95% CI [−0.01, 0.09]), indicating that we did not find statistically significant evidence regarding whether individuals' baseline verbal fluency levels were associated with differences in the linear rate of decline as hearing impairment worsened. In contrast, the covariance between the intercept and the quadratic slope was significantly negative (−0.27, 95% CI [−0.37, −0.18]), suggesting that individuals with higher baseline verbal fluency exhibited a less pronounced acceleration in decline than those with lower initial scores. Furthermore, the covariance between the linear and quadratic slopes was significantly positive (0.24, 95% CI [0.04, 0.45]), showing that individuals who had a steeper initial decline tended to experience a stronger accelerating deterioration over time. Together, these findings suggest that while we did not find statistically significant evidence regarding the association between baseline verbal fluency and variation in the linear decline due to hearing impairment, baseline performance may still be related to the degree of acceleration in decline over time, highlighting that non-linear effects of hearing loss on executive function may differ by initial performance levels.

### Comparison across cognitive domains

To address the second research question—whether the associations between social isolation/loneliness profiles and hearing impairment vary across cognitive domains—we examined three-way interactions between cognitive domain (i.e., immediate recall vs delayed recall and verbal fluency), social isolation/loneliness profiles, and hearing impairment (Table 5). Immediate recall was used as the reference category for cognitive domain, and all cognitive scores were z-standardized to facilitate comparison across domains (for the alternative reference category see Supplementary Table 5).

Analysis of the three-way interaction revealed significant differences in the associations between hearing impairment and cognitive performance across loneliness/isolation profiles. Overall, greater hearing impairment was associated with lower cognitive performance ($b = -0.02$, 95% CI [−0.03, −0.01]), and this association was stronger among individuals in the non-isolated but lonely profile ($b = -0.04$, 95% CI [−0.05, −0.02]). However, the strength of these associations varied across cognitive domains. Specifically, for individuals who were non-isolated but lonely, the relationship between hearing impairment and cognitive performance differed depending on the domain: the negative association with hearing impairment was attenuated for both verbal fluency ($b = 0.03$, 95% CI [0.01, 0.04]) and delayed recall ($b = 0.02$, 95% CI [0.0001, 0.04]) when compared to immediate recall. This indicates that the hearing-cognition association was strongest for immediate recall in this group. Notably, these domain-specific differences were observed only among the non-isolated but lonely individuals. These findings suggest that the relationship between hearing impairment and cognitive function may vary not only by psychosocial profile but also by the specific cognitive domain assessed, particularly among individuals experiencing loneliness in the absence of objective social isolation.

### Discussion

This study investigated whether the longitudinal association between subjective hearing impairment and cognitive outcomes varies depending on profiles of social isolation and loneliness, accounting for both interindividual differences and intraindividual change. Specifically, we examined the extent to which levels and changes of episodic memory and executive functioning were associated with subjective hearing impairment, and how

these associations differed across distinct psychosocial profiles. Our findings demonstrate that older adults in certain isolation/loneliness profiles—particularly those who were both socially isolated and lonely—showed poorer cognitive performance and steeper cognitive decline as hearing impairment worsened. These findings highlight the public health importance of identifying at-risk subgroups for accelerated cognitive aging.

Our results extend previous research showing that subjective hearing impairment is associated with both lower cognitive performance and faster decline[2–4,8,9]. We observed this pattern consistently across both episodic memory measures (immediate and delayed recall) and executive functioning (verbal fluency). Furthermore, the significant quadratic effects suggest that cognitive decline accelerates with greater hearing impairment, —supporting the view that progressive sensory loss may pose compounding risks to cognitive health in aging. Notably, we also found significant between-person differences across social isolation and loneliness profiles. Individuals in more socially disadvantaged profiles—particularly those who were both isolated and lonely—consistently performed worse across all cognitive domains compared to those in the non-isolated and not lonely profile. This supports the argument that both objective and subjective social disconnection are relevant to cognitive outcomes and may jointly contribute to risk.

A key contribution of our study is the identification of the lonely-in-the-crowd—individuals who are not isolated but still feel lonely—as an especially vulnerable group. This group showed a stronger negative association between increasing hearing impairment and memory decline than the non-lonely, non-isolated group. Unlike past cross-sectional studies that found limited evidence for moderation by loneliness[28], our longitudinal design uncovered a significant role for loneliness in shaping the extent to which sensory decline is linked to cognition. Importantly we found no statistically significant evidence for moderating effects for the quadratic or overall levels of hearing impairment, suggesting that associations with social and emotional factors may be limited to the linear component of the hearing-cognition relationship. Identifying psychosocial profiles that moderate the hearing impairment-cognition relationship offers new insights into differential risk patterns for cognitive decline.

With respect to executive functioning, the isolated and lonely profile showed the lowest performance overall, but we found no evidence that social isolation and/or loneliness moderated the relationship between hearing impairment and cognitive decline in this domain. Although the graphical trends pointed toward greater decline in this group, confidence intervals were wide, indicating heterogeneity and the need for larger samples in this specific profile to clarify these patterns. Future research should investigate in more depth whether executive functions are more or less sensitive to the psychosocial context compared to memory processes.

Our multivariate analyses further revealed a significant three-way interaction, suggesting that domain-specific associations between hearing impairment and cognition vary across psychosocial profiles. Among the lonely-in-the-crowd group, the negative associations between hearing impairment and both delayed recall and verbal fluency were attenuated relative to immediate recall, indicating that the association of hearing loss with cognitive outcomes may be more pronounced in certain cognitive domains. This domain-specific pattern was not observed in more socially isolated groups. These findings imply that loneliness may more strongly associate with the hearing-memory relationship than the hearing-executive function link. One explanation could be that memory processes—especially encoding and retrieval—are more closely tied to socially enriched contexts, making them more sensitive to psychosocial disruptions[46] while executive tasks like verbal fluency rely more heavily on internal strategies and less on social engagement[47]. Future research should explore the specific effects of hearing impairment and social isolation on different measures of executive functioning, such as attentional control, working memory, or reasoning, to better understand these domain-specific relationships.

Several potential mechanisms may underlie these observed associations. The cognitive load hypothesis[48,49] posits that hearing impairment could increase the mental effort required to process auditory input,

**Table 5 | Multivariate multilevel model with fixed and random effects (immediate recall as reference category)**

| Predictors | B (β) | SE | T (df) | CI | p |
|---|---|---|---|---|---|
| **Fixed effects** | | | | | |
| (Intercept) | 1.10 (0.04) | 0.03 | 35.50 (35,130) | 1.04 to 1.16 | <0.001 |
| Retest effects | | | | | |
| Wave 2 | 0.07 (0.02) | 0.01 | 11.31 (342,700) | 0.06 to 0.08 | <0.001 |
| Wave 4 | 0.12 (0.03) | 0.01 | 20.38 (338,200) | 0.11 to 0.13 | <0.001 |
| Wave 5 | 0.13 (0.04) | 0.01 | 25.58 (331,700) | 0.12 to 0.14 | <0.001 |
| Wave 6 | 0.13 (0.05) | 0.01 | 25.30 (319,600) | 0.12 to 0.14 | <0.001 |
| Wave 7 | 0.12 (0.03) | 0.01 | 15.20 (352,000) | 0.11 to 0.14 | <0.001 |
| Wave 8 | 0.11 (0.04) | 0.01 | 13.88 (317,400) | 0.10 to 0.13 | <0.001 |
| Wave 9 | 0.12 (0.04) | 0.01 | 12.25 (311,800) | 0.10 to 0.13 | <0.001 |
| Between-subjects' effects | | | | | |
| Age (M) | −0.03 (−0.21) | 0.0004 | −64.46 (34,230) | −0.03 to −0.02 | <0.001 |
| Sex (Female respondents) | 0.15 (0.07) | 0.01 | 22.58 (33,170) | 0.13 to 0.16 | <0.001 |
| Education | 0.20 (0.28) | 0.02 | 85.45 (33,200) | 0.19 to 0.20 | <0.001 |
| Chronic conditions (M) | −0.02 (−0.03) | 0.003 | −8.78 (33,840) | −0.03 to −0.02 | <0.001 |
| Hearing impairment (M) | −0.08 (−0.06) | 0.004 | −18.35 (33,760) | −0.01 to −0.07 | <0.001 |
| Cognitive domains | | | | | |
| Verbal fluency | 0.02 (0.02) | 0.004 | 6.82 (343,700) | 0.02 to 0.03 | <0.001 |
| Delayed recall | 0.001 (0.001) | 0.004 | 0.19 (343,600) | −0.01 to 0.01 | 0.852 |
| Profiles of social isolation and loneliness (ref: non-isolated and low loneliness) | | | | | |
| Non-isolated and high loneliness | −0.11 (−0.11) | 0.01 | −15.41 (50,540) | −0.12 to −0.10 | <0.001 |
| Isolated and low loneliness | −0.14 (−0.14) | 0.04 | −3.81 (53,760) | −0.20 to −0.07 | <0.001 |
| Isolated and high loneliness | −0.24 (−0.24) | 0.03 | −8.25 (53,320) | −0.30 to −0.18 | <0.001 |
| Within-subjects' effects | | | | | |
| Age (C) | −0.02 (−0.07) | 0.001 | −23.98 (312,600) | −0.02 to −0.02 | <0.001 |
| Chronic conditions (C) | −0.002 (−0.002) | 0.001 | −1.55 (286,200) | −0.01 to 0.001 | 0.120 |
| Hearing impairment (C) | −0.02 (−0.01) | 0.01 | −4.85 (73,730) | −0.03 to −0.01 | <0.001 |
| Hearing impairment ($C^2$) | −0.03 (−0.02) | 0.003 | −8.66 (6073) | −0.03 to −0.02 | <0.001 |
| Interaction effects | | | | | |
| Verbal Fluency* Non-isolated and high loneliness | −0.07 (−0.07) | 0.01 | 11.81 (343,700) | −0.10 to −0.06 | <0.001 |
| Delayed recall * Non-isolated and high loneliness | −0.002 (−0.002) | 0.01 | −0.38 (343,700) | −0.01 to 0.01 | 0.705 |
| Verbal Fluency * Isolated and low loneliness | 0.12 (0.12) | 0.03 | 4.15 (343,700) | 0.10 to 0.17 | <0.001 |
| Delayed recall * Isolated and low loneliness | 0.01 (0.01) | 0.03 | 0.18 (343,600) | −0.01 to 0.06 | 0.860 |
| Verbal Fluency * Isolated and high loneliness | 0.02 (0.02) | 0.02 | −0.92 (343,800) | −0.02 to 0.07 | 0.359 |
| Delayed recall * Isolated and high loneliness | −0.002 (−0.002) | 0.02 | −0.09 (343,600) | −0.10 to 0.04 | 0.932 |
| Verbal Fluency * Hearing impairment (C) | −0.002 (−0.001) | 0.01 | 0.27 (344,000) | −0.01 to 0.01 | 0.784 |
| Delayed recall * Hearing impairment (C) | −0.004 (−0.0003) | 0.01 | 0.27 (343,800) | −0.01 to 0.01 | 0.993 |
| Non-isolated and high loneliness * Hearing impairment (C) | −0.04 (−0.02) | 0.04 | −5.02 (71,520) | −0.05 to −0.02 | <0.001 |
| Isolated and low loneliness * Hearing impairment (C) | −0.002 (−0.001) | 0.04 | −0.04 (70,790) | −0.07 to 0.07 | 0.965 |
| Isolated and high loneliness * Hearing impairment (C) | −0.01 (−0.01) | 0.03 | −0.49 (72,740) | −0.07 to 0.04 | 0.627 |
| Verbal Fluency * Non-isolated and high loneliness * Hearing impairment (C) | 0.03 (0.02) | 0.01 | 2.96 (344,100) | 0.01 to 0.04 | 0.003 |
| Delayed recall * Non-isolated and high loneliness * Hearing impairment (C) | 0.02 (0.01) | 0.01 | 1.97 (343,800) | 0.0001 to 0.04 | 0.049 |
| Verbal Fluency * Isolated and low loneliness * Hearing impairment (C) | −0.04 (−0.02) | 0.04 | −0.89 (343,600) | −0.13 to 0.05 | 0.372 |
| Delayed recall * Isolated and low loneliness * Hearing impairment (C) | −0.01 (0.003) | 0.04 | −0.10 (343,600) | −0.09 to 0.08 | 0.917 |
| Verbal Fluency * Isolated and high loneliness * Hearing impairment (C) | −0.02 (−0.01) | 0.04 | −0.46 (343,900) | −0.09 to 0.05 | 0.643 |
| Delayed recall * Isolated and high loneliness * Hearing impairment (C) | −0.02 (−0.01) | 0.04 | −0.42 (343,600) | −0.09 to 0.06 | 0.672 |

| | Estimates | SD | CI | |
|---|---|---|---|---|
| **Random effects** | | | | |
| Residual variance | 0.50 | 0.71 | 0.71 to 0.71 | |
| Intercept (variance) | 0.29 | 0.54 | 0.54 to 0.56 | |
| Hearing Impairment (C) slope (variance) | 0.04 | 0.21 | 0.20 to 0.22 | |
| Hearing Impairment ($C^2$) slope (variance) | 0.02 | 0.14 | 0.13 to 0.15 | |

**Table 5 (continued) | Multivariate multilevel model with fixed and random effects (immediate recall as reference category)**

| | Estimates | SD | CI |
|---|---|---|---|
| Intercept*Hearing impairment (C) slope (covariance) | 0.08 | – | 0.05 to 0.11 |
| Intercept*Hearing impairment (C²) slope (covariance) | −0.25 | – | −0.29 to −0.21 |
| Hearing impairment (C) slope* Hearing impairment (C²) slope (covariance) | 0.04 | – | −0.03 to 0.11 |
| ICC | 0.38 | | |
| N | 33,731 | | |
| Observations | 411,075 | | |
| Marginal R²/Conditional R² | 0.194/0.499 | | |

The CIs in the random effects' variances correspond to their Standard Deviations, while in the random effects' covariances they correspond to the actual estimates.

*M* person-mean variable (between-subjects differences), *C* person-mean centred variable indicating the linear change, *C²* person-mean centred variable indicating the quadratic change, *SD* standard deviation, *CI* 95% confidence intervals, *Marginal R²* variance explained by fixed effects only, *Conditional R²* variance explained by fixed and random effects.

potentially diverting cognitive resources from other tasks such as memory and executive functioning. This additional cognitive demand may also contribute to social withdrawal, leading to fewer opportunities for stimulation. Alternatively, declining cognition itself may exacerbate sensory processing challenges, suggesting a bidirectional or complex relationship.

Additionally, loneliness has been related to chronic stress, elevated cortisol levels, poor health behaviours (e.g., poor diet, sleep disturbances), all of which associate to worse brain health[50–52]. Research shows that compared to their non-lonely peers, lonely individuals find everyday stressors more distressing[26], particularly when the stressor is hearing loss[27]. Thus, for lonely individuals, hearing impairment may contribute to cognitive decline not only via perceptual demand but also through increased psychological burden[5,53]. From a cognitive reserve perspective, social interaction is a key form of cognitive stimulation that promotes neural resilience[54,55]; when both hearing ability and social engagement decline, individuals may be especially vulnerable.

Our findings indicate that both hearing impairment and psychosocial factors such as loneliness and social isolation may be relevant to cognitive functioning in later life. Although causality cannot be inferred, the co-occurrence of hearing difficulties and social disconnection—objective or subjective—may be important to consider in research on cognitive aging. These results align with the view that cognitive health in later life is associated with multiple, interrelated factors. Future work may benefit from examining how sensory, emotional, and social domains are linked to cognitive outcomes over time, particularly in the context of settings where real-life interventions are implemented.

## Limitations
Despite its strengths, this study has several limitations. First, while we opted to model social isolation and loneliness as distinct psychosocial profiles, we acknowledge that a factorial decomposition approach—modeling isolation and loneliness as separate factors with main and interaction effects—could provide complementary insights. Such an approach allows for the examination of both additive and interactive contributions of these constructs. However, we chose the profile-based strategy to better reflect real-world patterns of co-occurrence and to align with previous work conceptualizing social disconnection as a multidimensional experience. Future research could benefit from directly comparing profile-based and factorial approaches to assess how analytic choices influence conclusions about the joint effects of isolation and loneliness. Second, although this study tested the interplay of longitudinal hearing impairment and cognition with social isolation and loneliness profiles, one limitation is that the assessment of the psychosocial factors is cross-sectional. SHARE provides the first concurrent assessment of the social connectedness scale and loneliness in wave 6, followed by inconsistent assessment in the following waves. Future work should examine the dynamic associations of loneliness and isolation to cognition, providing also a distinction between those who are chronically socially isolated and/or lonely. Moreover, this study cannot infer causality, highlighting the need for longitudinal assessments to disentangle causal relationships between loneliness, isolation, hearing impairment, and cognition. As it often occurs in longitudinal studies, we acknowledge that we may have selective attrition, and respondents who dropped out

may differ systematically from those who remained. A sensitivity analysis comparing respondents who completed every wave to those who dropped out after the first wave revealed notable differences: those who dropped out were, on average, six years older, reported 0.5 more chronic illnesses, had 0.2 points worse hearing impairment, and demonstrated lower cognitive scores across all domains compared to those who remained. However, no differences were observed in educational attainment. While these differences highlight the potential impact of selective dropout on our findings, our approach of using complete cases mitigates some biases by ensuring consistent data for the analysis. Nonetheless, the potential underestimation of cognitive decline in more vulnerable groups warrants further investigation in future studies. Another limitation is that our final model did not include interaction terms between the primary predictors and demographic variables such as age, sex, or education. While this is a common modeling choice aimed at parsimony and interpretability, it entails the assumption that the associations between hearing impairment, psychosocial profiles, and cognitive outcomes are constant across these demographic subgroups. We explored a subset of potential interactions (e.g., with mean age) during preliminary analyses, but found no consistent evidence to support their inclusion in the final model. Nevertheless, the possibility of differential effects by sex, age, or education remains important, and future studies with a specific focus on subgroup variation may be better suited to test these moderation effects explicitly. Additionally, although SHARE includes data from multiple countries, we did not model country-specific random slopes due to the relatively small number of clusters (12 countries), which falls below the recommended threshold (20 – 30 clusters) for reliable estimation of higher-level random effects[56,57]. Future research with a larger number of country-level units or theoretical hypotheses regarding cross-national variation could explore these effects further. Another potential venue for future research may be the further investigation of these relationships in other cognitive domains, such as working memory, as an ubiquitous cognitive process in daily life that is known to decline with advancing age[58,59]. Regarding hearing impairment, a limitation of this study may be that SHARE only provides subjective hearing impairment information. Studying these relationships with an objective measure of hearing impairment (audiometry) could be another venue for future research. Lastly, a limitation of this study is that we offer insights for the European context exclusively. Future studies should aim to replicate these findings across cultural contexts, as the experience of social isolation and loneliness in older age may differ significantly by region.

## Conclusion
In conclusion, our study reveals critical insights regarding the interplay between hearing impairment, social isolation, loneliness and cognitive aging. We found that hearing impairment was associated with steeper decline in both episodic memory and executive functioning, and that this association was stronger among individuals reporting higher levels of subjective loneliness, even in the absence of objective social isolation. These associations were stronger for episodic memory compared to executive function. Our study underscores the importance of a holistic approach that combines auditory health with psychosocial support to maintain cognitive health in later life.

## Data availability
The data are freely accessible upon written request. The data can be accessed through the SHARE project website—www.share-project.org.

## Code availability
The software code (R 4.4.2)[60] associated with this publication is accessible here: https://zenodo.org/records/15620371.

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

## Acknowledgements
This work was supported by the Swiss National Centre of Competence in Research LIVES—Overcoming vulnerability: Life course perspectives, which is financed by the Swiss National Science Foundation (grant number: 51NF40-185901). The authors are grateful to the Swiss National Science Foundation for its financial assistance. The Swiss National Science Foundation had no role in study design, data collection and analysis, decision to publish or preparation of the manuscript.

## Author contributions
C.L. conceived the study, performed the statistical analysis and drafted the first version of the manuscript, while she revised the second version and the final version of the manuscript. S.Z. drafted the first version of the manuscript and revised the following versions of the manuscript. N.T. drafted the first version of the manuscript and revised the following versions of the manuscript. E.J.-B. provided statistical advice and revised all versions of the manuscript. M.M. revised all versions of the manuscript. G.L. revised all versions of the manuscript. C.S. revised all versions of the manuscript. A.R. revised all versions of the manuscript. M.K. revised all versions of the manuscript. A.I. conceived the study, supervised the execution of the project and revised all versions of the manuscript. All authors have read and approved the final version of the manuscript and agree with the order of presentation of the authors.

## Competing interests
The authors declare no competing interests.
