## [Transparent Peer Review file · Communications Psychology]

Profiles of Social Isolation and Loneliness as Moderators of the Longitudinal Association Between Uncorrected Hearing Impairment and Cognitive Aging

Corresponding Author: Dr Charikleia Lampraki

Version 0:

Decision Letter:

Dear Dr Lampraki,

Thank you for your patience during the peer-review process. Your manuscript titled "Profiles of Social Isolation and Loneliness as Moderators of the Longitudinal Association Between Uncorrected Hearing Impairment and Cognitive Aging" has now been seen by 2 reviewers, whose comments are appended below. You will see that they find your work of some potential interest. However, they have raised quite substantial concerns that must be addressed. In light of these comments, we cannot accept the manuscript for publication, but would be interested in considering a revised version that fully addresses these serious concerns.

We hope you will find the Reviewers' comments useful as you decide how to proceed. Should additional work allow you to address these criticisms, we would be happy to look at a substantially revised manuscript. If you choose to take up this option, please highlight all changes in the manuscript text file, and provide a detailed point-by-point reply to the reviewers. The revision should also address the comments regarding clarity in the reporting of results and links to the literature.

The reviewers' concerns around the applied modeling approach, in particular the dichotomization of variables, modeling of interaction effects, nesting of effects within countries, test-retest effects should be addressed in a revision by providing corresponding statistical, not just narrative, evidence for the findings. A revision should also address comments regarding the clarity and interpretation of the results as well as relations to the existing literature around social isolation and loneliness.

I am attaching a checklist that details critical reporting requirements for the revised manuscript. Please attend to each item and ensure your manuscript is fully compliant. We are requesting that your manuscript aligns with these requirements as this facilitates the evaluation of your manuscript, reducing delays in re-review and potential future acceptance. If your revised manuscript is not aligned with these requests on major issues, such as those concerning statistics, it may be returned to you for further revisions without re-review. Additional information can be found in our style and formatting guide Communications Psychology formatting guide.

If the revision process takes significantly longer than five months, we will be happy to reconsider your paper at a later date, provided it still presents a significant contribution to the literature at that stage.

Please use the following link to submit your

- revised manuscript,
- point-by-point response to the referees' comments,
- cover letter (as a separate document),
- the Editorial Policy Checklist (see below),
- the Reporting Summary (see below), and
- the completed Editorial Request Table (attached):

Link Redacted

Thank you for the opportunity to review your work.

Best regards,

Yana Fandakova

Yana Fandakova
External Editor
Communications Psychology

REVIEWER EXPERTISE:

Reviewer #1 cognitive and psychosocial aging, longitudinal modeling

Reviewer #2 cognitive aging and longitudinal modeling

REVIEWER REPORTS:

Reviewer #1 (Remarks to the Author):

The authors used longitudinal data from SHARE to examine whether social isolation and loneliness moderate the association between hearing impairment and cognitive function. Between persons, they found that being lonely and/or socially isolated was related to lower cognitive performance. Within persons, being not isolated but lonely was related to lower performance in Immediate and Delayed Recall Tests relative to being not isolated and not lonely. The manuscript addresses an interesting and relevant topic, but there are several limitations and areas where clarity must be improved:

(1) The theoretical rationale for examining moderating effects of social isolation and loneliness is not well-developed. As a possible explanation, the authors note: "Notably, both social isolation and loneliness have been related to hearing impairment which, when combined with social isolation, could accelerate cognitive decline due to a lack of mental stimulation, and may in addition increase frustration and mental fatigue, which loneliness can exacerbate.", but there is no reference to relevant theoretical frameworks, supporting literature, and the dynamics of the moderation are not clarified.

(2) Similarly, the rationale for examining the interaction between loneliness and social isolation is also not clear. This issue is briefly touched upon on page 6 for the first time, but a theoretical rational linking possible combinations of social interactions of loneliness and social isolation to hearing loss and cognitive decline is missing.

(3) A major concern is the dichotomization of the social isolation and loneliness variables. This leads to considerable information loss. The research questions outlined on page 7 could be addressed without dichotomizing, for example, by modeling interaction effects.

(4) On page 4, the authors draw upon the socio-emotional selectivity theory to argue that social isolation may be a conscious choice. According to the socio-emotional selectivity theory, the conscious choice relates to the prioritization of emotionally meaningful social relationships, not of social isolation.

(5) Some of the analyses performed are not mentioned previously or justified in the manuscript, for example, quadratic effects of hearing impairment or comparisons between European regions.

Reviewer #2 (Remarks to the Author):

The current manuscript can be seen as an interesting extension of previous work investigating the associations of hearing impairment, cognition and loneliness/isolation. Importantly, the authors broaden the scope on cognition by investigating episodic memory and executive functions and they investigate the multivariate associations of hearing impairment, cognition, and isolation using longitudinal data. The topic is interesting and the work provides novel aspects to answering

the question. However, my initial enthusiasm for the manuscript was considerably dampened by some concerns regarding the model and the interpretation of its result. Please see my detailed points below.

Major points:

- * The mixed effects model ignores an important nesting structure of the SHARE study. Since participants are nested within countries and there is reason to assume between-country differences in the effects of interest, this nested data structure should be reflected in the mixed model with appropriate country-specific random effects.
- * Even though the authors list several possible causal pathways (e.g., sensory decline -> cognition, or loneliness -> less stimulation -> poorer cognition), the study is an observational study, which decomposes longitudinal associations but cannot be used to derive strong causal claims. In some parts, the manuscript uses strong causal language, which is not supported by the model (e.g., "social isolation and loneliness profiles (e.g., non-isolated but lonely) exacerbate the relationship between hearing impairment and cognition" when really we can only say that particular profiles are associated with stronger or weaker associations of hearing and cognition). For example, all observed effects are always be consistent with unobserved third variables that cause hearing loss, loss of cognition, and/or isolation/loneliness.
- * The four profiles were dummy-coded in the mixed model. However, it's a perfect two-factorial design, so wouldn't it be appropriate to code both main effects and the interaction within the mixed-model? Then, one could statistically identify both the additive as well as the super/supra-additive effect of isolation and loneliness. For example, Figure 1 seems to mostly show two main effects but no interaction whereas Figure 2 may be due to an interaction (or noise - we don't really now without the statistics).
- * I find Figure 1-2 very helpful as illustration of the model predictions based on the profiles. However, wouldn't it make sense to illustrate the quadratic effect given that the model is quadratic?
- * While the model answers the first research question, the second research question is not addressed using any statistical test or model comparison: " 2) Are potential differences between the four loneliness and social isolation profiles consistent across cognitive abilities (i.e., episodic memory, executive functions)?"
- * The present analysis assumes no test/retest effects in any of the cognitive tasks, which I find hard to believe. I suggest adding a test-retest predictor. For example, Ghisletta et al. (2014) tried both a first-exposure dummy coding as well as a linear exposure model (counting the number of previous exposures to the task) and determined the best by model comparison.
- * The results section is a mere qualitative summary and I miss a discussion based on standardized and unstandardized effect sizes (as partly shown in Table 1). Interpretations based on significance are mostly useful in a sample the size of the SHARE data. Here, it is only suprising if something is non-significant -- given the Crud factor, our expectations is that any effect is going to be significant with 30k+ individuals. For example, going from excellent hearing (1) to poor (5) is a maximal detrimental change of 4 units in hearing impairment. Looking at Table 1, I see that the linear effect of hearing impairment on verbal fluency is -0.25, that is, this maximum loss of hearing is associated with only $-0.25 \cdot 4 = -1$, so this person can name only a single animal less after this severe hearing loss (or did I miss something?). I urge the authors to thoroughly revise their result section based on actual effect sizes and their practical relevance. Specifically because the authors promise "clinically valuable insights" (p.14).

Minor points:

- Unless the age distribution is normal in the sample, it could be informative to give robust estimates of dispersion, such as median and quantiles) instead of or in addition to mean and SD.
- was education median-split across or within countries?
- potential inconsistency in the dichotomization of loneliness and isolation. Loneliness was dichotomized using a median split whereas isolation was dichotomized using a extreme group split (0 vs others). Why not both median or both extreme group? How sensitive are the results to this choice?
- what was the rationale for dichotomizing education instead of adding it as a linear effect?
- adding standardized effect size estimates would be quite useful for Table 1 since predictors have quite different scales; it's hard to compare their relative influence on the prediction as it is
- The resolution of the plots is quite low. If you allow me to joke, the line drawing somewhat reminded me of a TRS-80. In all seriousness, would it be possible to use shaded regions to illustrate standard errors of these estimates around the lines?
- Are the other positive that all interaction effects of all predictors are zero?

This is a signed review,
Andreas Brandmaier

References:

Ghisletta, P., Bäckman, L., Bertram, L., Brandmaier, A. M., Gerstorf, D., Liu, T., & Lindenberger, U. (2014). The Val/Met polymorphism of the brain-derived neurotrophic factor (BDNF) gene predicts decline in perceptual speed in older adults. *Psychology and Aging, 29*(2), 384–392. <https://doi.org/10.1037/a0035201>

EDITORIAL POLICIES

We ask that you ensure your manuscript complies with our editorial policies and reporting requirements.

To that end, we require revised manuscripts to be accompanied by two completed items: a reporting summary that collects information on study design and procedure, and an editorial policy checklist that verifies compliance with all required editorial policies

- <https://www.nature.com/documents/nr-reporting-summary.zip>>Nature Research Reporting Summary
- <https://www.nature.com/documents/nr-editorial-policy-checklist.pdf>>Editorial Policy Checklist

All points on the policy checklist must be addressed. Your revised manuscript can only be sent back to the referees if these checklists are completed and uploaded with the revision.

Notes: If you have submitted a Stage 1 Registered Report, Review, Primer, Comment, or Perspective you do not need to submit these forms. If you have already submitted these forms, you may disregard this request.

** Visit Nature Research's author and referees' website at <http://www.nature.com/authors>>www.nature.com/authors for information about policies, services and author benefits**

Communications Psychology is committed to improving transparency in authorship. As part of our efforts in this direction, we are now requesting that all authors identified as 'corresponding author' create and link their Open Researcher and Contributor Identifier (ORCID) with their account on the Manuscript Tracking System prior to acceptance. ORCID helps the scientific community achieve unambiguous attribution of all scholarly contributions. You can create and link your ORCID from the home page of the Manuscript Tracking System by clicking on 'Modify my Springer Nature account' and following the instructions in the link below. Please also inform all co-authors that they can add their ORCID to their accounts and that they must do so prior to acceptance.

If you experience problems in linking your ORCID, please contact the <http://platformsupport.nature.com/>>Platform Support Helpdesk.

Version 1:

Decision Letter:

Dear Dr Lampraki,

Your manuscript titled "Profiles of Social Isolation and Loneliness as Moderators of the Longitudinal Association Between Uncorrected Hearing Impairment and Cognitive Aging" has now been seen by our reviewers, whose comments appear below. In light of their advice I am delighted to say that we are happy, in principle, to publish a suitably revised version in Communications Psychology.

We therefore invite you to revise your paper one last time to address the remaining concerns of our reviewers and a list of editorial requests. At the same time we ask that you edit your manuscript to comply with our format requirements and to maximise the accessibility and therefore the impact of your work.

EDITORIAL REQUESTS:

SUBMISSION INFORMATION:

OPEN ACCESS:

* DATA AVAILABILITY:

Link Redacted

Best regards,

Jennifer Bellingtier

Jennifer Bellingtier, PhD
Senior Editor
Communications Psychology

Yana Fandakova
External Editor
Communications Psychology

REVIEWER EXPERTISE:

Reviewer #1 cognitive and psychosocial aging, longitudinal modeling
Reviewer #2 cognitive aging and longitudinal modeling

REVIEWERS' COMMENTS:

Reviewer #1 (Remarks to the Author):

The authors were responsive to issues raised by the reviewers and the manuscript is much improved.

While I appreciate the authors' clarification of the reason for dichotomizing loneliness and social isolation, I disagree with the statement that social isolation is inherently discrete (page 22). Conceptually and empirically, there can be varying degrees of social isolation.

The histograms included in the revision letter indicate that social isolation and loneliness as defined by the authors were rather rare. It would be good to include sample sizes for the four distinct profiles.

Reviewer #2 (Remarks to the Author):

I thank the reviewers for thoroughly addressing all of my concerns. Reading the response letter was highly informative, and I appreciate the detailed clarification of the authors' modeling choices. In particular, the explanation regarding effect sizes was especially helpful in deepening my understanding of the modeling results. I also apologize for one of my earlier comments being non-sensical (item 13 in the response letter). I believe I meant to ask how the authors justify that all interactions between the chosen predictors are zero. Of course, setting interactions to zero is usually the default modeling choice striving for parsimony, however, the model implicitly assumes that all effects (other than a mean difference in the outcome) are constant across age, sex and education. Regarding the discussion of distinct profiles versus a factorial decomposition (item 3): Decomposing the influences of isolation and loneliness using a factorial design does not restrict the analysis to additive effects of both. Importantly, it would allow to separate a potential linear additive component (main effect) from a sub/super-additive component (interaction). I still think, this decomposition is a worthwhile perspective but I accept the authors' justification that the distinct profiles better reflect their conceptual thinking and better align with previous research.

This is a signed review,
Andreas Brandmaier

Dear Editor,

Thank you very much for the opportunity to revise and resubmit our manuscript titled: “Profiles of Social Isolation and Loneliness as Moderators of the Longitudinal Association Between Uncorrected Hearing Impairment and Cognitive Aging.”

We are grateful to you and the two reviewers for the constructive and insightful feedback. We have thoroughly revised the manuscript in an extensive way to respond to all major and minor comments, and we believe that the revised version is now considerably strengthened—both conceptually and methodologically.

In the following paragraphs we summarize the most substantial revisions made (followed by a detailed point-to-point response to all reviewer comments below this letter):

- Clarified theoretical rationale and hypotheses: We restructured the introduction to provide a clearer and more coherent rationale for testing the moderating role of social isolation and loneliness, grounding our approach in relevant theoretical frameworks and supporting literature.
- Statistical modelling improvements and transparency:
 - We included retest markers to account for learning effects across waves using dummy-coded variables for prior test exposures, following the reviewer’s recommendation and relevant literature (e.g., Ghisletta et al., 2014).
 - We conducted additional analyses to account for country-level nesting and provide two supplementary tables (in the response) showing models with both individual- and country-level intercepts, and with both intercepts and slopes for the reviewer to inspect.
 - We included a new multivariate model to address the second research question—whether domain-specific cognitive outcomes differ in their associations with hearing impairment and social isolation/loneliness profiles. These results are presented in Table 2 of the manuscript and discussed accordingly.
- Refined interpretation of findings:
 - We revised the results and discussion to emphasize effect sizes and practical relevance over statistical significance, given the large sample size.
 - We carefully revised all causal language and replaced it with terminology that reflects the observational nature of our study.
 - We now provide clear, practical interpretations of our findings.
- Justification for modelling choices:
 - We clarified our rationale for using predefined social profiles rather than factorial interaction models, citing conceptual reasoning, and precedent in the literature (e.g., Menec et al., 2020; Zaccaria et al., 2022).
- Presentation and clarity improvements:

- We improved the resolution and interpretability of Figures 1–2 and added shaded confidence intervals.
- We included new supplementary figures showing quadratic effects of hearing impairment for each cognitive outcome.
- Tables now include both standardized and unstandardized coefficients, with a clear explanation of our standardization approach.
- Expanded limitations section: We now explicitly discuss the absence of country-level random slopes due to the limited number of clusters (12 countries). This addition reflects our commitment to transparency and the reviewers’ emphasis on statistical rigor.

We have provided a detailed point-to-point response to all reviewer comments and highlighted changes throughout the revised manuscript as requested. We also attach the Editorial Policy Checklist, Reporting Summary, and Editorial Request Table.

We hope you find the revised manuscript suitable for re-review. Please do not hesitate to reach out should further information be needed.

Thank you again for your time and consideration.

Warm regards,

Charikleia Lampraki, PhD

Point-to-point response to reviewers’ comments

Reviewer #1 (Remarks to the Author):

The authors used longitudinal data from SHARE to examine whether social isolation and loneliness moderate the association between hearing impairment and cognitive function. Between persons, they found that being lonely and/or socially isolated was related to lower cognitive performance. Within persons, being not isolated but lonely was related to lower performance in Immediate and Delayed Recall Tests relative to being not isolated and not lonely. The manuscript addresses an interesting and relevant topic, but there are several limitations and areas where clarity must be improved:

Thank you very much for taking the time to review our manuscript. We have thoroughly addressed all your comments and concerns, and you can find our detailed replies below.

(1) The theoretical rationale for examining moderating effects of social isolation and loneliness is not well-developed. As a possible explanation, the authors note: “Notably, both social isolation and loneliness have been related to hearing impairment which, when combined with social isolation, could accelerate cognitive decline due to a lack of mental stimulation, and may in addition increase frustration and mental fatigue, which loneliness can exacerbate.”, but there is no reference to relevant theoretical frameworks, supporting literature, and the dynamics of the

moderation are not clarified.

Thank you very much for your comment. We realise that the theoretical justification for conducting a moderation analysis was underdeveloped in the previous version of the manuscript. We have now restructured the introduction and better developed our rationale based on relevant theoretical frameworks, adding new references, and explaining how we expect the association between worsening hearing impairment and cognition to vary by profiles of social isolation and loneliness. We are convinced that these amendments will better support our rationale and analysis. Pp. 3-6

(2) Similarly, the rationale for examining the interaction between loneliness and social isolation is also not clear. This issue is briefly touched upon on page 6 for the first time, but a theoretical rationale linking possible combinations of social interactions of loneliness and social isolation to hearing loss and cognitive decline is missing.

Thank you again for mentioning that the rationale for the interaction between social isolation and loneliness as well as its potential moderating effect on the link between hearing impairment and cognition was not well developed in the previous version of the manuscript. In line with our response to your previous comment, we have restructured and further developed the rationale for our analysis, supporting it with theoretical frameworks and previous empirical findings (e.g., Menec et al., 2020). We hope that you are now more satisfied with the current version of the introduction.

(3) A major concern is the dichotomization of the social isolation and loneliness variables. This leads to considerable information loss. The research questions outlined on page 7 could be addressed without dichotomizing, for example, by modelling interaction effects.

Thank you for your thoughtful comment. There are both conceptual and empirical reasons why we chose to dichotomize the two variables and use the derived profiles, rather than testing their interaction term or modelling them as continuous variables. First, from an empirical standpoint, loneliness is highly skewed in our data (see graph below). Including it as a continuous variable would introduce bias in estimation and interpretation. A median split ensures a more balanced distribution of participants across groups, enhancing statistical power and allowing us to differentiate between individuals experiencing higher versus lower loneliness in a stable and interpretable way. While alternative classification methods (e.g., tertiles, quartiles) were considered, our chosen method provided the most conceptually coherent and statistically robust grouping. In contrast, social isolation is inherently discrete and often treated categorically in prior research—for example, by distinguishing between living alone versus living with others. We followed this approach in treating isolation dichotomously, which also reflects meaningful differences in social connectedness.

Second, conceptually, our approach is based on the idea that combinations of loneliness and social isolation represent qualitatively distinct social experiences—rather than simply additive or multiplicative effects of two variables. While it is possible to model these dimensions using an interaction term (e.g., lonely × isolated), we argue that this structure imposes symmetry and linearity assumptions that are unlikely to reflect the true psychological and social nuances of these states. For instance, someone who is socially isolated but not lonely (e.g., content living

alone) is fundamentally different from someone who is lonely but not socially isolated (e.g., surrounded by people but emotionally disconnected). These differences may not be well captured by an interaction model that assumes a continuous or synergistic relationship between variables. By using a dummy-coded profile approach, we aim to capture these ecologically meaningful and theory-driven social categories—an approach supported by prior literature that treats these combinations as distinct constructs with differential associations to health and cognition (e.g., Menec et al., 2020; Steptoe et al., 2013; Zaccaria et al., 2022). This allows us to directly test hypotheses about the unique effects of each psychosocial profile, without constraining the analysis to a particular functional form (e.g., additive or multiplicative). In doing so, we address the risk of information loss not by avoiding dichotomization altogether, but by ensuring that the grouping strategy reflects both the statistical characteristics of the data and the theoretical goals of the study. We have clarified these methodological decisions in the revised manuscript to enhance transparency. We thank the reviewer again for raising this important point. P.22

(4) On page 4, the authors draw upon the socio-emotional selectivity theory to argue that social isolation may be a conscious choice. According to the socio-emotional selectivity theory, the conscious choice relates to the prioritization of emotionally meaningful social relationships, not of social isolation.

Thank you for your comment. We agree that mentioning socioemotional selectivity theory could cause confusion, and after careful consideration we decided to not refer to socioemotional selectivity theory anymore. The updated version of the manuscript has a clearer narrative regarding isolation and loneliness.

(5) Some of the analyses performed are not mentioned previously or justified in the manuscript, for example, quadratic effects of hearing impairment or comparisons between European regions.

Thank you for your suggestion. We have now mentioned in the introduction that worsening of hearing impairment has been related in the past with accelerated cognitive decline, justifying the investigation of quadratic (non-linear) effects of hearing impairment in our analysis. Regarding the analysis between European regions, we did this analysis in an exploratory way, as we did not have specific hypotheses linked to the different regions of Europe, but because it could be informative as SHARE is a multinational study and the question of between-countries differences is often raised. If we had a larger number of countries, we would have nested the individuals into countries, adding an extra level in our multilevel models. For the reasons mentioned above (e.g., small number of countries) and below (in the reply to the first comment of the second reviewer) we were not able to do so, and, therefore, decided not to include this

exploratory result in the results section, but mention it as a limitation – the fact that we were not able to add country-level random slopes – and propose that future research may focus on differences across countries. Pp. 5 and 18

Reviewer #2 (Remarks to the Author):

The current manuscript can be seen as an interesting extension of previous work investigating the associations of hearing impairment, cognition and loneliness/isolation. Importantly, the authors broaden the scope on cognition by investigating episodic memory and executive functions and they investigate the multivariate associations of hearing impairment, cognition, and isolation using longitudinal data. The topic is interesting and the work provides novel aspects to answering the question. However, my initial enthusiasm for the manuscript was considerably dampened by some concerns regarding the model and the interpretation of its result. Please see my detailed points below.

Thank you very much for taking the time to review our manuscript. We have thoroughly addressed all your comments and concerns and provide our detailed replies below.

Major points:

1) The mixed effects model ignores an important nesting structure of the SHARE study. Since participants are nested within countries and there is reason to assume between-country differences in the effects of interest, this nested data structure should be reflected in the mixed model with appropriate country-specific random effects.

We appreciate the reviewer's insightful suggestion regarding the inclusion of country-specific random effects to account for the nesting structure within the SHARE study. We acknowledge that there may be between-country differences in cognitive aging and hearing impairment due to variations in healthcare access, cultural factors, and social policies. However, after careful consideration, we decided not to include country-specific random slopes in our model. The main reason is that we only have a relatively small number of countries (12) at the higher level, while the recommendations suggest at least 20-30 clusters (e.g., Maas & Hox, 2005; McNeish & Stapleton, 2016) in order to have reliable estimations of the random slopes. Second, we did not have country-specific hypotheses and introducing country-specific random slopes would significantly increase model complexity, making interpretation more challenging without a clear theoretical justification for country-level variation in the effects of interest. As we want to be transparent and provide you with sufficient information, we also provide at the end of this letter two tables: 1) with models that include country-specific and individual random intercepts and 2) with models that include both country-specific and individual random intercepts and slopes.

For the reasons mentioned above, we believe that the current models, presented in Table 1 of the manuscript, have a balance between statistical rigor and interpretability while remaining consistent with our theoretical framework. We appreciate the reviewer's suggestion and have carefully considered its implications, but we respectfully maintain that the inclusion of country-level random slopes is not essential for answering our research questions. However, we decided to include this limitation in our discussion. P. 18

Maas, C. J., & Hox, J. J. (2005). Sufficient sample sizes for multilevel modeling. *Methodology*, 1(3), 86–92.

McNeish, D. & Stapleton, L. M. (2016). The effect of small sample size on two-level model estimates: A review and illustration. *Educational Psychology Review*, 28(2), 295–314.

2) Even though the authors list several possible causal pathways (e.g., sensory decline -> cognition, or loneliness -> less stimulation -> poorer cognition), the study is an observational study, which decomposes longitudinal associations but cannot be used to derive strong causal claims. In some parts, the manuscript uses strong causal language, which is not supported by the model (e.g., "social isolation and loneliness profiles (e.g., non-isolated but lonely) exacerbate the relationship between hearing impairment and cognition" when really we can only say that particular profiles are associated with stronger or weaker associations of hearing and cognition). For example, all observed effects are always be consistent with unobserved third variables that cause hearing loss, loss of cognition, and/or isolation/loneliness.

Thank you very much for this comment. We acknowledge that this is an observational study and have tried to avoid causal language. However, we understand that some parts may have indicated otherwise. We have therefore carefully revised the introduction and the rest of the manuscript, and we hope that the reviewer will be happy with the rephrasing of previous potentially causal associations.

3) The four profiles were dummy-coded in the mixed model. However, it's a perfect two-factorial design, so wouldn't it be appropriate to code both main effects and the interaction within the mixed-model? Then, one could statistically identify both the additive as well as the super/supra-additive effect of isolation and loneliness. For example, Figure 1 seems to mostly show two main effects but no interaction whereas Figure 2 may be due to an interaction (or noise - we don't really now without the statistics).

Thank you for this insightful suggestion. While the structure of the four loneliness/isolation profiles could be mathematically expressed as a 2x2 factorial design (socially isolated vs. not x lonely vs. not), we argue that a factorial coding does not align with the conceptual goals of our study. Our approach is based on the premise that combinations of loneliness and social isolation represent qualitatively distinct social experiences—not merely additive effects of two separate constructs. For example, individuals who are socially isolated but not lonely (e.g., those who live alone but feel emotionally fulfilled) likely differ meaningfully from those who are lonely but not isolated (e.g., surrounded by others but feeling emotionally disconnected). Treating these groups as interaction terms in a factorial model imposes a structure that assumes symmetry or linearity in how these dimensions combine, which may obscure the distinctiveness of these profiles in practice. Moreover, we chose a dummy-coded profile approach to reflect prior literature that treats these combinations as emergent and interpretable social categories with potentially different implications for cognition and health (e.g., Menec et al., 2020; Steptoe et al., 2013; Zaccaria et al., 2022). This approach allows us to test theory-driven hypotheses about the specific nature of each profile—rather than estimating an interaction term that assumes a particular functional form (e.g., multiplicative or additive). Finally, from a statistical perspective, the assumption behind a factorial interaction model is that the combination of loneliness and

isolation exerts synergistic or suppressor effects. However, the effect of, say, social isolation on cognition may not be conditional on loneliness in a linear way (and vice versa). Profiles offer greater interpretability by capturing non-additive, non-linear, and ecologically meaningful groupings that correspond to real-world social experiences.

4) I find Figure 1-2 very helpful as illustration of the model predictions based on the profiles. However, wouldn't it make sense to illustrate the quadratic effect given that the model is quadratic?

Thank you for your suggestion. Figures 1-2 show the interaction between the change in hearing impairment and the profiles of social isolation and loneliness. However, as you can also see in Table 1, only the linear effect of hearing impairment interacted significantly with the profiles. The interaction with the quadratic term was not improving the model and therefore excluded from the final solution (parsimony). Therefore, although the main effects of the linear and the quadratic terms of hearing impairment are significant, only the linear term is interacting with the profiles and is depicted in the graphs. However, based on one of your comments below we improved the quality of the graphs and added the shaded regions to illustrate standard errors. In addition, we created three new Figures (A, B and C) that we included in the Supplementary Material depicting the hearing impairment linear and quadratic trends of the fixed effects for each cognitive outcome. We hope that our response and amendments clarify any confusion about the figures. Pp. 35-36

5) While the model answers the first research question, the second research question is not addressed using any statistical test or model comparison: " 2) Are potential differences between the four loneliness and social isolation profiles consistent across cognitive abilities (i.e., episodic memory, executive functions)?

Thank you for your important comment. We agree that in the previous version of the manuscript we had not included a proper statistical analysis comparing whether the effects we found in the separate cognitive domains statistically differed. Therefore, in the new version we have included a multivariate model where the type of cognitive domains is included as a factor and the cognitive scores are z-standardized, to facilitate the comparison. Using a triple interaction between cognitive domains*loneliness/isolation profiles*hearing impairment (change), we found that there are differences between the memory tasks and the executive functioning task. We have included an extra table (Table 2) and amended accordingly the results and discussion sections of the manuscript. Pp.14-15, 16-17, 24, 33-34

6) The present analysis assumes no test/retest effects in any of the cognitive tasks, which I find hard to believe. I suggest adding a test-retest predictor. For example, Ghisletta et al. (2014) tried both a first-exposure dummy coding as well as a linear exposure model (counting the number of previous exposures to the task) and determined the best by model comparison.

Thank you for suggesting this approach, which we had not initially considered. Following your recommendation and the methodology proposed by Ghisletta et al. (2014), we tested two models: one including binary variables indicating whether the participant had previously taken

the test at each wave (except the first one which is the reference), and another using a continuous variable counting the total number of previous test exposures.

Consistent with the findings of Ghisletta et al. (2014), model comparison (AIC/BIC) indicated that the model with binary test-retest variables provided a better fit. The test-retest effects were statistically significant across all cognitive tasks, with performance improving with each repeated exposure (e.g., a 0.07-word increase in Immediate Recall at wave 2, gradually rising to a 0.23-word increase by wave 5). This confirms the presence of practice effects in the cognitive measures.

We have retained this model in the manuscript and updated the methods and results sections accordingly. We appreciate the reviewer's insightful suggestion, which has strengthened the robustness of our analysis. PP. 21, 31-32 and the presentation of results e.g., p.10

7) The results section is a mere qualitative summary and I miss a discussion based on standardized and unstandardized effect sizes (as partly shown in Table 1). Interpretations based on significance are mostly useful in a sample the size of the SHARE data. Here, it is only surprising if something is non-significant -- given the Crud factor, our expectations is that any effect is going to be significant with 30k+ individuals. For example, going from excellent hearing (1) to poor (5) is a maximal detrimental change of 4 units in hearing impairment. Looking at Table 1, I see that the linear effect of hearing impairment on verbal fluency is -0.25, that is, this maximum loss of hearing is associated with only $-0.25 \times 4 = -1$, so this person can name only a single animal less after this severe hearing loss (or did I miss something?). I urge the authors to thoroughly revise their result section based on actual effect sizes and their practical relevance. Specifically because the authors promise "clinically valuable insights" (p.14).

Thank you for your valuable feedback regarding the interpretation of our results. We agree that with large samples like those in the SHARE dataset ($N > 30,000$), statistical significance alone provides limited insight, and a thorough discussion of effect sizes is essential to understand practical relevance.

We revised our results section to emphasize unstandardized effect sizes and their practical implications. As you correctly calculated, a change from excellent hearing (1) to poor hearing (5) was associated with approximately one fewer animal named in the verbal fluency task (4-unit increase in hearing impairment \times -0.24 coefficient = -0.96). We have now incorporated this type of interpretation throughout our results. Regarding the standardized coefficients, we used the `effectsize` function from the R package `effectsize` (Ben-Shachar et al., 2020) with the "refit" method to calculate them (Tables 1 and 2), acknowledging the ongoing debate regarding standardization in multilevel models (e.g., Wang et al., 2019). We chose this approach to balance the need for interpretable coefficients while respecting the complex multilevel structure of our data.

Our revised results section now includes:

1. Explicit interpretations of what these effects mean in real-world terms. For example, we now explain that "an individual who experiences a 4-unit decline in hearing ability (e.g., going from excellent to poor hearing) would recall, on average, 1.00 fewer words on the verbal fluency task."

2. Contextualization of these effects within meaningful frameworks, such as clarifying that an average individual with a baseline delayed recall score of 4.1 (on a 1-10 scale) could expect a reduction of approximately 0.80 points over a decade due to the combined effects of aging, hearing impairment, and social factors, representing approximately 20% of their baseline cognitive score.
3. Cumulative impact assessments rather than isolated effects. For instance, we now highlight that for individuals in the "lonely in the crowd" profile, a 4-unit decline in hearing would contribute to a 0.20-point additional decline in recall scores, which combines with age-related decline (0.40 points per decade) and the direct effect of hearing impairment.
4. Clearer presentation of the clinical relevance of our findings, showing how seemingly small individual effects can accumulate to substantial real-world impacts over time, particularly when considering long-term trajectories of cognitive aging.

We believe these revisions provide a more balanced view of our findings that accurately reflect both the statistical and practical significance of the relationships we observed. We have adjusted our discussion of "clinically valuable insights", following the recommendations of the journal about proposing interventions or policy changes when not directly testing them . Pp. 7-15

Ben-Shachar MS, Lüdtke D, & Makowski D (2020). "effectsize: Estimation of Effect Size Indices and Standardized Parameters." *Journal of Open Source Software*, 5(56), 2815.
doi:10.21105/joss.02815.

Wang, L., Zhang, Q., Maxwell, S. E., & Bergeman, C. S. (2019). On Standardizing Within-Person Effects: Potential Problems of Global Standardization. *Multivariate Behavioral Research*, 54(3), 382–403. <https://doi.org/10.1080/00273171.2018.1532280>

Minor points:

8) Unless the age distribution is normal in the sample, it could be informative to give robust estimates of dispersion, such as median and quantiles) instead of or in addition to mean and SD.

Thank you for your insightful comment. We acknowledge that mean and standard deviation alone may not fully capture the age distribution, particularly if it is not normally distributed. To provide a more robust description of age dispersion, we have now included the interquartile range (IQR: 54–67) alongside the median (60), mean (61.4), standard deviation (8.6), and age range (50–99 years) in the methods section of the manuscript. We believe this revision offers a more comprehensive representation of the age distribution in our sample. P.20

9) what was the rationale for dichotomizing education instead of adding it as a linear effect? Was education median-split across or within countries?

Thank you for this comment. The rationale for including education as a dichotomous variable was only simplicity of interpretation. However, we acknowledge that having the continuous score may improve the model. Therefore, we now included education as a continuous score instead of the dichotomous predictor. The interpretation of the results did not change with this replacement. In the previous version of the manuscript, where we had used the dichotomous variable, the education was split across countries. P. 21

10) potential inconsistency in the dichotomization of loneliness and isolation. Loneliness was dichotomized using a median split whereas isolation was dichotomized using an extreme group split (0 vs others). Why not both median or both extreme group? How sensitive are the results to this choice?

Thank you again for considering the implications regarding the dichotomization of the two variables. First, loneliness is highly skewed in our data (see graph below), a common finding in social psychology research, where individuals often report not feeling lonely. A median split ensures a more balanced distribution of participants across groups, enhancing statistical power and allowing us to differentiate between individuals who are not lonely versus lonely in a stable and interpretable way. While alternative classification methods (e.g., tertiles, quartiles) were considered, our chosen method provided the most conceptually coherent and statistically robust grouping. In contrast, social isolation is inherently discrete and often treated categorically in prior research—for example, by distinguishing between living alone versus living with others. The measure of social connectedness used is a very rich measure providing information about quantitative and qualitative aspects of connectedness. When someone has a score of 0, it really reflects an individual who quantitatively and qualitatively is isolated, and we argue that their situation differs from other individuals who have some level of connectedness. We followed this approach in treating isolation dichotomously, reflecting meaningful differences in social connectedness. P.22

11) Adding standardized effect size estimates would be quite useful for Table 1 since predictors have quite different scales; it's hard to compare their relative influence on the prediction as it is.

We appreciate the reviewer's suggestion regarding standardized effect size estimates to aid interpretability. In response, we have added standardized estimates to Table 1 using the effectsize package in R with the refit method, as recommended for multilevel models. This allows for a general sense of the relative magnitude of effects across predictors on different scales. However, we have opted not to emphasize or interpret the standardized coefficients in the main text. As mentioned also above, there is ongoing debate in the methodological literature about the appropriateness and interpretability of standardized coefficients in mixed-effects models, particularly in the presence of random slopes and hierarchical structures (e.g., rights and implications of standardizing within vs. between clusters). Given these complexities, we present the standardized estimates as a supplementary aid for readers but focus our interpretation on the unstandardized effects, which are more directly tied to the original metrics of interest. Pp. 31-34

12) The resolution of the plots is quite low. If you allow me to joke, the line drawing somewhat reminded me of a TRS-80. In all seriousness, would it be possible to use shaded regions to illustrate standard errors of these estimates around the lines?

Thank you for your suggestion. Indeed, the quality was not good in the previous version. We have updated the plots accordingly. Pp. 36 and 37

13) Are the other positive that all interaction effects of all predictors are zero?

We apologize but we cannot be sure of what you meant with this comment. If we may guess, you are asking whether the other possible interaction effects were non-significant. As described in the methods section, we included each interaction term individually in the model and assessed whether the fit improved based on several fit indices. The models we present are the ones with the best fit and parsimony, including the interactions that improved the overall fit.

Thank you again for your thorough review of the manuscript and your helpful remarks and suggestions to improve!

Table A. Multilevel models with fixed and random effects (individual- and country-level intercepts)

Predictors	Immediate recall			Delayed recall			Verbal fluency		
	B (β)	CI	p	B (β)	CI	p	B (β)	CI	p
Fixed Effects									
(Intercept)	7.85 (0.02)	7.63 – 8.07	<0.001	6.18 (0.01)	5.87 – 6.48	<0.001	28.53 (0.02)	26.75 – 30.31	<0.001
Retest Effects									
Wave 2	0.08 (0.01)	0.05 – 0.12	<0.001	0.34 (0.02)	0.29 – 0.38	<0.001	0.37 (0.02)	0.21 – 0.53	<0.001
Wave 4	0.24 (0.03)	0.21 – 0.27	<0.001	0.69 (0.05)	0.63 – 0.76	<0.001	0.87 (0.02)	0.65 – 1.09	<0.001
Wave 5	0.22 (0.04)	0.19 – 0.24	<0.001	0.83 (0.05)	0.75 – 0.90	<0.001	1.60 (0.04)	1.35 – 1.86	<0.001
Wave 6	0.21 (0.05)	0.18 – 0.24	<0.001	1.03 (0.07)	0.94 – 1.11	<0.001	1.95 (0.07)	1.66 – 2.25	<0.001
Wave 7	0.25 (0.04)	0.21 – 0.29	<0.001	1.07 (0.04)	0.97 – 1.17	<0.001	2.51 (0.05)	2.15 – 2.88	<0.001
Wave 8	0.19 (0.04)	0.14 – 0.23	<0.001	1.17 (0.04)	1.06 – 1.29	<0.001	3.29 (0.09)	2.89 – 3.70	<0.001
Wave 9	0.22 (0.04)	0.17 – 0.27	<0.001	1.29 (0.05)	1.16 – 1.41	<0.001	3.68 (0.10)	3.23 – 4.14	<0.001
Between-subjects' effects									
Age (M)	-0.05 (-0.24)	-0.05 – -0.05	<0.001	-0.06 (-0.25)	-0.07 – -0.06	<0.001	-0.18 (-0.19)	-0.19 – -0.17	<0.001
Sex (Females)	0.29 (0.08)	0.26 – 0.31	<0.001	0.41 (0.10)	0.38 – 0.45	<0.001	0.05 (0.003)	-0.06 – 0.16	0.377
Education (High)	0.27 (0.23)	0.27 – 0.28	<0.001	0.32 (0.22)	0.31 – 0.33	<0.001	1.13 (0.21)	1.09 – 1.17	<0.001
Chronic conditions (M)	-0.06 (-0.04)	-0.07 – -0.05	<0.001	-0.07 (-0.04)	-0.09 – -0.06	<0.001	-0.21 (-0.03)	-0.25 – -0.16	<0.001
Hearing impairment (M)	-0.13 (-0.06)	-0.14 – -0.11	<0.001	-0.12 (-0.04)	-0.14 – -0.10	<0.001	-0.44 (-0.04)	-0.51 – -0.36	<0.001

Profiles of social isolation and loneliness (ref: non-isolated and low loneliness)

Non-isolated and high loneliness	-0.16 (-0.10)	-0.19 – -0.14	<0.001	-0.16 (-0.08)	-0.19 – -0.13	<0.001	-0.65 (-0.08)	-0.77 – -0.54	<0.001
Isolated and low loneliness	-0.27 (-0.16)	-0.38 – -0.15	<0.001	-0.23 (-0.11)	-0.38 – -0.08	0.003	-0.59 (-0.08)	-1.13 – -0.05	0.033
Isolated and high loneliness	-0.38 (-0.22)	-0.48 – -0.29	<0.001	-0.36 (-0.17)	-0.48 – -0.24	<0.001	-1.34 (-0.17)	-1.79 – -0.90	<0.001
Within-subjects' effects									
Age (C)	-0.04 (-0.09)	-0.04 – -0.03	<0.001	-0.04 (-0.08)	-0.05 – -0.04	<0.001	-0.24 (-0.13)	-0.25 – -0.22	<0.001
Chronic conditions (C)	-0.01 (-0.01)	-0.02 – -0.00	0.021	-0.01 (-0.01)	-0.02 – -0.00	0.006	0.03 (0.004)	0.00 – 0.06	0.026
Hearing impairment (C)	-0.03 (-0.02)	-0.05 – -0.02	<0.001	-0.05 (-0.01)	-0.06 – -0.03	<0.001	-0.25 (-0.02)	-0.29 – -0.21	<0.001
Hearing impairment (C ²)	-0.05 (-0.02)	-0.06 – -0.04	<0.001	-0.05 (-0.01)	-0.07 – -0.04	<0.001	-0.22 (-0.02)	-0.27 – -0.18	<0.001
Interaction effects									
Hearing impairment (C) * Non-isolated and high loneliness	-0.06 (-0.02)	-0.08 – -0.04	<0.001	-0.04 (-0.01)	-0.07 – -0.02	0.001	-	-	-
Hearing impairment (C) * Isolated and low loneliness	0.02 (0.01)	-0.08 – 0.13	0.681	0.03 (0.01)	-0.10 – 0.15	0.684	-	-	-
Hearing impairment (C) * Isolated and high loneliness	-0.04 (-0.01)	-0.12 – 0.05	0.400	-0.08 (-0.02)	-0.19 – 0.02	0.103	-	-	-

Random Effects

Residual variance	1.50 (1.23)	2.08 (1.44)	22.75 (4.78)
Intercept (individual-level variance)	0.71 (0.84)	1.29 (1.13)	18.62 (4.32)
Intercept (country-level variance)	0.11 (0.32)	0.20 (0.44)	8.53 (2.95)
ICC	0.35	0.42	0.54
N participants	33726	33725	33727
N countries	12	12	12
Observations	137005	137039	137031
Marginal R ² / Conditional R ²	0.185 / 0.474	0.177 / 0.521	0.134 / 0.605

Table B. Multilevel model with fixed and random effects (individual- and country-level intercepts and slopes) for outcomes of interest

Predictors	Immediate recall			Delayed recall			Verbal fluency		
	B (β)	CI	p	B (β)	CI	p	B (β)	CI	p
Fixed Effects									
(Intercept)	7.85 (0.02)	7.63 – 8.07	<0.001	7.00 (0.01)	6.70 – 7.29	<0.001	29.74 (0.02)	27.99 – 31.4 9	<0.001
Retest Effects									
Wave 2	0.08 (0.01)	0.04 – 0.11	<0.001	0.16 (0.02)	0.12 – 0.20	<0.001	0.55 (0.02)	0.42 – 0.69	<0.001
Wave 4	0.24 (0.03)	0.21 – 0.27	<0.001	0.39 (0.05)	0.35 – 0.42	<0.001	0.52 (0.02)	0.39 – 0.64	<0.001
Wave 5	0.22 (0.04)	0.19 – 0.24	<0.001	0.32 (0.05)	0.29 – 0.36	<0.001	1.04 (0.04)	0.93 – 1.15	<0.001
Wave 6	0.21 (0.05)	0.18 – 0.24	<0.001	0.35 (0.07)	0.32 – 0.39	<0.001	1.28 (0.07)	1.17 – 1.39	<0.001
Wave 7	0.25 (0.04)	0.21 – 0.29	<0.001	0.33 (0.04)	0.28 – 0.39	<0.001	1.69 (0.05)	1.52 – 1.87	<0.001
Wave 8	0.18 (0.03)	0.14 – 0.22	<0.001	0.28 (0.04)	0.23 – 0.33	<0.001	2.19 (0.09)	2.02 – 2.36	<0.001

Wave 9	0.22 (0.04)	0.17 – 0.27	<0.001	0.31 (0.05)	0.25 – 0.37	<0.001	2.47 (0.10)	2.27 – 2.66	<0.001
Between-subjects' effects									
Age (M)	-0.05 (-0.24)	-0.05 – -0.05	<0.001	-0.06 (-0.25)	-0.07 – -0.06	<0.001	-0.18 (-0.19)	-0.19 – -0.17	<0.001
Sex (Females)	0.29 (0.08)	0.26 – 0.31	<0.001	0.41 (0.10)	0.38 – 0.44	<0.001	0.05 (0.003)	-0.06 – 0.16	0.370
Education (High)	0.27 (0.23)	0.27 – 0.28	<0.001	0.32 (0.22)	0.31 – 0.33	<0.001	1.13 (0.21)	1.09 – 1.17	<0.001
Chronic conditions (M)	-0.06 (-0.04)	-0.07 – -0.05	<0.001	-0.07 (-0.04)	-0.09 – -0.06	<0.001	-0.21(-0.03)	-0.25 – -0.16	<0.001
Hearing impairment (M)	-0.12 (-0.06)	-0.14 – -0.11	<0.001	-0.12 (-0.04)	-0.14 – -0.10	<0.001	-0.44 (-0.04)	-0.51 – -0.36	<0.001
Profiles of social isolation and loneliness (ref: non-isolated and low loneliness)									
Non-isolated and high loneliness	-0.16 (-0.10)	-0.19 – -0.14	<0.001	-0.16 -(0.08)	-0.19 – -0.13	<0.001	-0.65 (-0.08)	-0.77 – -0.54	<0.001
Isolated and low loneliness	-0.27 (-0.16)	-0.38 – -0.15	<0.001	-0.22 (-0.11)	-0.37 – -0.07	0.003	-0.59 (-0.08)	-1.13 – -0.05	0.032
Isolated and high loneliness	-0.38 (-0.22)	-0.47 – -0.28	<0.001	-0.36 (-0.17)	-0.48 – -0.24	<0.001	-1.34 (-0.17)	-1.79 – -0.90	<0.001

Within-subjects effects

Age (C)	-0.04 (-0.09)	-0.04 – -0.03	<0.001	-0.04 (-0.08)	-0.04 – -0.04	<0.001	-0.23 (-0.13)	-0.25 – -0.22	<0.001
Chronic conditions (C)	-0.01 (-0.004)	-0.02 – -0.00	0.035	-0.01 (-0.01)	-0.02 – -0.01	0.021	0.04 (0.004)	0.01 – 0.06	0.021
Hearing impairment (C)	-0.04 (-0.02)	-0.08 – -0.00	0.034	-0.06 (-0.02)	-0.11 – -0.02	0.008	-0.24 (-0.02)	-0.34 – -0.13	<0.001
Hearing impairment (C ²)	-0.05 (-0.02)	-0.07 – -0.04	<0.001	-0.05 (-0.02)	-0.08 – -0.03	<0.001	-0.23 (-0.02)	-0.35 – -0.12	<0.001

Interaction effects

Hearing impairment (C) * Non-isolated and high loneliness	-0.05 (-0.02)	-0.07 – -0.02	<0.001	-0.03 (-0.01)	-0.06 – -0.00	0.047			
Hearing impairment (C) * Isolated and low loneliness	0.01 (0.002)	-0.11 – 0.12	0.931	0.01 (-0.001)	-0.13 – 0.14	0.928			
Hearing impairment (C) * Isolated and high loneliness	-0.02 (-0.01)	-0.11 – 0.08	0.709	-0.06 (-0.02)	-0.18 – 0.05	0.249			

Estimates (SD)

CI

Estimates (SD)

CI

Estimates (SD)

CI

Random Effects

Residual variance	1.45 (1.21)	2.01 (1.42)	22.05 (4.70)
Intercept (individual variance)	0.74 (0.86)	1.34 (1.16)	19.20 (4.38)
Intercept (country variance)	0.11 (0.33)	0.21 (0.46)	8.66 (2.94)
Hearing Impairment (C) slope (individual-level variance)	0.08 (0.28)	0.11 (0.33)	1.18 (1.09)
Hearing Impairment (C ²) slope (individual-level variance)	0.01 (0.12)	0.02 (0.14)	0.33 (0.57)
Hearing Impairment (C) slope (country-level variance)	0.04 (0.07)	0.01 (0.07)	0.03 (0.16)
Hearing Impairment (C ²) slope (country-level variance)	0.001 (0.02)	0.001 (0.04)	0.03 (0.18)
Intercept*Hearing impairment (C) slope (individual-level; covariance)	0.07	0.10	0.02
Intercept*Hearing impairment (C ²) slope (individual-level; covariance)	-0.26	-0.30	-0.26

Intercept*Hearing impairment (C) slope (country-level; covariance)	0.56	0.71	0.27
Intercept*Hearing impairment (C ²) slope (country-level; covariance)	-0.19	-0.94	-0.42
Hearing impairment (C) slope* Hearing impairment (C ²) slope (individual-level; covariance)	-.13	-0.32	0.25
Hearing impairment (C) slope* Hearing impairment (C ²) slope (country-level; covariance)	.61	-0.46	-0.54
ICC	0.37	0.44	0.56
N participants	33726	33725	33727
N countries	12	12	12
Observations	137005	137039	137031
Marginal R ² / Conditional R ²	0.185 / 0.489	0.177 / 0.536	0.133 / 0.617

REVIEWERS' COMMENTS:

Reviewer #1 (Remarks to the Author):

*The authors were responsive to issues raised by the reviewers and the manuscript is much improved.

Thank you again for taking the time to review our manuscript and propose solutions that improved its quality and clarity!

*While I appreciate the authors' clarification of the reason for dichotomizing loneliness and social isolation, I disagree with the statement that social isolation is inherently discrete (page 22). Conceptually and empirically, there can be varying degrees of social isolation.

Thank you very much for pointing this out. You are correct that social isolation can have different levels. However, as the social connectedness measure assesses different aspects of connectedness (or isolation) we wanted our binary indicator to reflect the contrast between complete isolation versus any level of connectedness. What we meant with our previous wording is that “no contact” has a substantive salience. Steptoe et al. (2013) found, for instance, that the most isolated individuals had significantly higher mortality probability, suggesting that this level of isolation can be used as a high-risk threshold in public-health. We have now added this clarification in p.9 of the manuscript.

Steptoe, A., Shankar, A., Demakakos, P., & Wardle, J. (2013). *Social isolation, loneliness, and all-cause mortality in older men and women*. *Proceedings of the National Academy of Sciences of the United States of America*, 110(15), 5797–5801.
<https://doi.org/10.1073/pnas.1219686110>

You can find below the altered text: . Although social isolation can vary in degree and is sometimes treated as a continuous construct, it is often operationalized categorically in prior research—for example, by distinguishing between living alone versus living with others. Following this precedent and given our interest in identifying individuals at the most extreme end of the isolation spectrum, we created a binary indicator based on the social connectedness scale: individuals scoring 0 (indicating no social connections) were coded as 1 (isolated), and those scoring 1 or higher were coded as 0 (not isolated).

*The histograms included in the revision letter indicate that social isolation and loneliness as defined by the authors were rather rare. It would be good to include sample sizes for the four distinct profiles.

Thank you for your comment. We have included the sample sizes for the four profiles in Table 1 (former Table A in the supplementary material) with all the descriptives for the study variables.

Reviewer #2 (Remarks to the Author):

*I thank the reviewers for thoroughly addressing all of my concerns. Reading the response letter was highly informative, and I appreciate the detailed clarification of the authors' modeling choices. In particular, the explanation regarding effect sizes was especially helpful in deepening my understanding of the modeling results.

Thank you again for taking the time to review our manuscript and for the useful suggestions and comments!

*I also apologize for one of my earlier comments being non-sensical (item 13 in the response letter). I believe I meant to ask how the authors justify that all interactions between the chosen predictors are zero. Of course, setting interactions to zero is usually the default modeling choice striving for parsimony, however, the model implicitly assumes that all effects (other than a mean difference in the outcome) are constant across age, sex and education.

Thank you for the clarification regarding your earlier comment (item 13), and we appreciate your thoughtful suggestions regarding our modeling approach. You are absolutely right that, by not including certain interaction terms in the final model, we implicitly assume that the effects of predictors are constant across subgroups such as age, sex, and education. Our decision to exclude interactions was primarily driven by parsimony, as you noted, as well as by concerns about model complexity and interpretability. We did consider potential interactions—particularly between change in hearing impairment and age (mean) or sex—during model development, but preliminary analyses did not reveal compelling evidence for meaningful (significantly improving the fit) interaction effects that would warrant inclusion. Incorporating all possible interactions would have substantially increased the number of parameters and raised the risk of overfitting. Moreover, these interaction effects were not central to the primary research questions of this study. That said, we agree that the assumption of effect homogeneity is important to make explicit, and we have now clarified this in the revised manuscript (p.24 in Discussion). We also note that exploring possible moderation by sex, education, or even chronic conditions may be a valuable direction for future research focused specifically on subgroup differences.

Here is the amendment to the discussion part: Another limitation is that our final model did not include interaction terms between the primary predictors and demographic variables such as age, sex, or education. While this is a common modeling choice aimed at parsimony and interpretability, it entails the assumption that the associations between hearing impairment, psychosocial profiles, and cognitive outcomes are constant across these demographic subgroups. We explored a subset of potential interactions (e.g., with mean age) during preliminary analyses, but found no consistent evidence to support their inclusion in the final model. Nevertheless, the possibility of differential effects by sex, age, or education remains important, and future studies with a specific focus on subgroup variation may be better suited to test these moderation effects explicitly.

*Regarding the discussion of distinct profiles versus a factorial decomposition (item 3): Decomposing the influences of isolation and loneliness using a factorial design does not restrict the analysis to additive effects of both. Importantly, it would allow to separate a potential linear additive component (main effect) from a sub/super-additive component (interaction). I still think, this decomposition is a worthwhile perspective but I accept the authors' justification that the distinct profiles better reflect their conceptual thinking and better align with previous research.

Thank you very much for acknowledging our conceptual justification. As noted, our decision to use distinct psychosocial profiles was guided by theoretical considerations and aligns with prior research that conceptualizes social disconnection as a multidimensional experience. We appreciate the reviewer's recognition of this rationale and have now added a note in the Discussion (p. 23) acknowledging the value of a factorial decomposition as a complementary

analytical approach for future research.

We added the following text: First, while we opted to model social isolation and loneliness as distinct psychosocial profiles, we acknowledge that a factorial decomposition approach— modeling isolation and loneliness as separate factors with main and interaction effects— could provide complementary insights. Such an approach allows for the examination of both additive and interactive contributions of these constructs. However, we chose the profile-based strategy to better reflect real-world patterns of co-occurrence and to align with previous work conceptualizing social disconnection as a multidimensional experience. Future research could benefit from directly comparing profile-based and factorial approaches to assess how analytic choices influence conclusions about the joint effects of isolation and loneliness.

Thank you again for your thoughtful suggestions that have improved the clarity of our manuscript.